# Reverse engineering synthetic antiviral amyloids

Emiel Michiels [1,2,10], Kenny Roose [3,4,10], Rodrigo Gallardo [1,2], Ladan Khodaparast[1,2], Laleh Khodaparast[1,2], Rob van der Kant [1,2], Maxime Siemons[1,2,5], Bert Houben [1,2], Meine Ramakers[1,2], Hannah Wilkinson[1,2], Patricia Guerreiro[1,2], Nikolaos Louros[1,2], Suzanne J. F. Kaptein [6], Lorena Itatí Ibañez [3,4], Anouk Smet[3,4], Pieter Baatsen[1,7], Shu Liu[8], Ina Vorberg [8,9], Guy Bormans[5], Johan Neyts[6], Xavier Saelens [3,4], Frederic Rousseau [1,2✉] & Joost Schymkowitz [1,2✉]

Human amyloids have been shown to interact with viruses and interfere with viral replication. Based on this observation, we employed a synthetic biology approach in which we engineered virus-specific amyloids against influenza A and Zika proteins. Each amyloid shares a homologous aggregation-prone fragment with a specific viral target protein. For influenza we demonstrate that a designer amyloid against PB2 accumulates in influenza A-infected tissue in vivo. Moreover, this amyloid acts specifically against influenza A and its common PB2 polymorphisms, but not influenza B, which lacks the homologous fragment. Our model amyloid demonstrates that the sequence specificity of amyloid interactions has the capacity to tune amyloid-virus interactions while allowing for the flexibility to maintain activity on evolutionary diverging variants.

[1] VIB Center for Brain and Disease Research, Leuven, Belgium. [2] Switch Laboratory, Department of Cellular and Molecular Medicine, KU Leuven, Leuven, Belgium. [3] VIB Center for Medical Biotechnology, Ghent, Belgium. [4] Department of Biomedical Molecular Biology, Ghent University, Ghent, Belgium. [5] Laboratory for Radiopharmacy, Department of Pharmaceutical and Pharmacological Sciences, KU Leuven, Leuven, Belgium. [6] Laboratory of Virology and Chemotherapy, Rega Institute for Medical Research, Department of Microbiology, Immunology and Transplantation, KU Leuven, Leuven, Belgium. [7] Electron Microscopy Platform of VIB Bio Imaging Core, KU Leuven, Leuven, Belgium. [8] German Center for Neurodegenerative Diseases (DZNE e.V.), Bonn, Germany. [9] Rheinische Friedrich-Wilhelms-Universität Bonn, Bonn, Germany. [10]These authors contributed equally: Emiel Michiels, Kenny Roose. ✉email: frederic.rousseau@kuleuven.vib.be; joost.schymkowitz@kuleuven.vib.be

For decades, human amyloids were mainly studied as the major hallmarks of protein deposition in disease, but in recent years functional amyloids, which have been uncovered in several human tissues[1] and throughout all kingdoms of life, have caused a revision of the view that amyloid structures are intrinsically toxic or pathological. Recently, increasing evidence supports the hypothesis that even disease-associated amyloids may not only be pathogenic byproducts[2], but that they could in fact also possess functional benefits for instance acting as interactors of human pathogens, including viruses[3–6]. For example, it has been suggested that the Alzheimer's disease (AD)-associated peptide, amyloid beta (Aβ), may have a role in innate immunity against herpes virus brain infections[5]. More specifically, Eimer et al.[5] showed that Aβ interacts with herpes glycoproteins, thereby inducing Aβ amyloid formation. The seeding of Aβ aggregation leads to an entrapment of the viral particles, which results in an amyloid-driven antiviral effect. Whether reciprocally the amyloid deposits formed as a result of such encounters with virus have a role in the development of AD remains to be determined, but herpes infection is a known risk factor for AD[7].

Aggregation seeding, as observed for Aβ when binding to herpes virus, is a well-described process, driven by so-called aggregation-prone regions (APRs)[8]. These APRs engage into homotypic interactions to form tightly packed intermolecular amyloid structures, resulting in a highly sequence-specific process[9–12], but allowing the interaction between proteins sharing highly homologous APRs. Interestingly, Aβ also shares a homolog APR sequence with the envelope glycoprotein B[4], a herpes protein for which an interaction with Aβ has already been shown[5]. To study whether an amyloid that shares a homolog APR with a certain viral protein can indeed specifically interact with this protein and thereby interfere with viral replication, we resorted to synthetic biology approach.

We designed two synthetic amyloids, each encoding a virus-specific APR identified in influenza A and Zika virus (ZIKV) proteins, respectively. We found that each amyloid interferes with the replication of its corresponding virus, without any cross-reactivity. For influenza A we show that our synthetic amyloid accumulates at the site of infection and interferes with influenza A replication in a murine infection model. The amyloid binds to its specific viral target protein in an APR-specific manner, forcing the protein into a nonfunctional conformation. Influenza B, which lacks the homolog APR, is not affected by the synthetic amyloid, highlighting the sequence specificity of this interaction. Together, we present the design of fully synthetic amyloids, solely based on an APR identified in a viral protein, that show specific antiviral properties. Moreover, our results demonstrate that APR-mediated amyloid interactions possess the same characteristics defining bona fide functional biological interactions including specificity and in vivo selectivity.

## Results

### Design of an APR-based synthetic peptide with antiviral properties.
To investigate the potential for amyloids to interact with viruses encoding a homologous APR, we here used a synthetic biology approach, using influenza A as a model virus. To design synthetic amyloids that share a homologous APR with an influenza A protein, we first identified all APRs in the influenza A proteome using the statistical thermodynamics algorithm TANGO[13]. We identified 40 APRs with a minimum TANGO score of 5 in the influenza A/Puerto Rico/8/34 (A/PR8) proteome. Synthetic peptides were designed based on those APRs using a tandem repeat scaffold[14–16]: two identical APRs, supercharged with arginine residues and linked by two amino acids (Fig. 1a, Supplementary Table 1). The tandem design is used to stimulate

amyloid formation of our synthetic peptides, while at the same it increases avidity of APR interactions. The charged amino acids, on the other hand, are added to decrease the kinetics of self-aggregation. These amyloid-prone peptides were screened for antiviral activity against two influenza A strains: the H1N1 strain A/PR8 and the H5N1 strain A/NIBRG-14 (Supplementary Fig. 1a, b). Madin–Darby canine kidney (MDCK) cells were treated with peptide at a concentration of 10 μM, 2 h prior to infection. Following overnight incubation, the amount of newly produced virus was determined by TCID50 titration. Overall, peptide 12, derived from an APR identified in the cap-binding domain of polymerase basic protein 2 (PB2), $_{381}$LIQLIVS$_{387}$ (Fig. 1b), showed the most pronounced effect. Importantly, the observed antiviral effect did not originate from aspecific cytotoxic effects (Supplementary Fig. 1c). We selected the APR of peptide 12, $_{381}$LIQLIVS$_{387}$, as the primary target APR and optimized the peptide design to maximize antiviral activity and minimize hemolytic activity. Different peptide variants were designed: multiple charged amino acids and linker residues were screened and mutations were inserted inside the APR to increase solubility while retaining antiviral activity (Supplementary Table 2). The antiviral effect of all peptide variants was screened by a plaque-size reduction assay and a hemolytic activity assay was performed on human red blood cells (Supplementary Fig. 1d, e). Taking both parameters into account, the best scoring peptide sequence is WDLIQLIVSDGSDLIQLIVSD, referred to as peptide 12B, in which the target APR ($_{381}$LIQLIVS$_{387}$) is flanked by negatively charged aspartate residues and an N-terminal tryptophan is added to allow for accurate concentration determination. This plaque-size reduction assay showed an ~80% reduction of influenza A/PR8 plaque area when cells were treated with peptide 12B (Fig. 1c, d). The antiviral effect was compared with three known anti-influenza compounds: nucleozin, a nucleoprotein-targeting inhibitor[17], Tamiflu, a neuraminidase inhibitor, and VX-787, a PB2 inhibitor. Of note, positively charged variants of peptide 12B (12E–12H) show a similar antiviral effect in the plaque-size reduction assay but were not selected for further analysis because of the aspecific toxic effects on human red blood cells (Supplementary Fig. 1e). A dose–response experiment, in which influenza A/PR8-infected MDCK cells were treated with a peptide concentration ranging from 10 nM to 75 μM, allowed to deduce an IC$_{50}$ of ~1.79 μM for peptide 12B (Fig. 1e). The cytotoxic dose (TD$_{50}$ = 727 μM), derived from the hemolytic activity (Fig. 1e), and the IC$_{50}$ allowed us to estimate a therapeutic index (TI = TD$_{50}$/IC$_{50}$) for peptide 12B of 406. Moreover, no aspecific toxic effects were observed on MDCK and Human Embryonic Kidney (HEK 293T) cells with concentrations up to 250 μM (Fig. 1f and Supplementary Fig. 1f). Finally, we used two additional assays to validate the antiviral effect of peptide 12B: a multicycle replication assay and the protection against cytopathic effects (Supplementary Fig. 1g, h). Both assays confirm that peptide 12B acts as an antiviral peptide against influenza A.

### Peptide 12B shows antiviral activity in an in vivo infection model.
To determine whether peptide 12B retains its antiviral activity in vivo, cohorts of BALB/c mice were infected with influenza A/PR8 (2x LD$_{50}$) and treated daily with buffer, 2.5 mg/kg peptide 12B (both intravenously (i.v.)), or 25 mg/kg Tamiflu (oral gavage). After 6 days, a small but significant reduction in virus titers in lung homogenates was observed for 12B-treated mice compared with buffer-treated and Tamiflu-treated mice[18] (Fig. 1g). To control for any possible aspecific toxic effects of our synthetic peptide, we performed an in vivo toxicity study. BALB/c mice received daily i.v. injections for 2 weeks with 10 mg/kg peptide 12B or buffer. Blood analysis and a histopathological

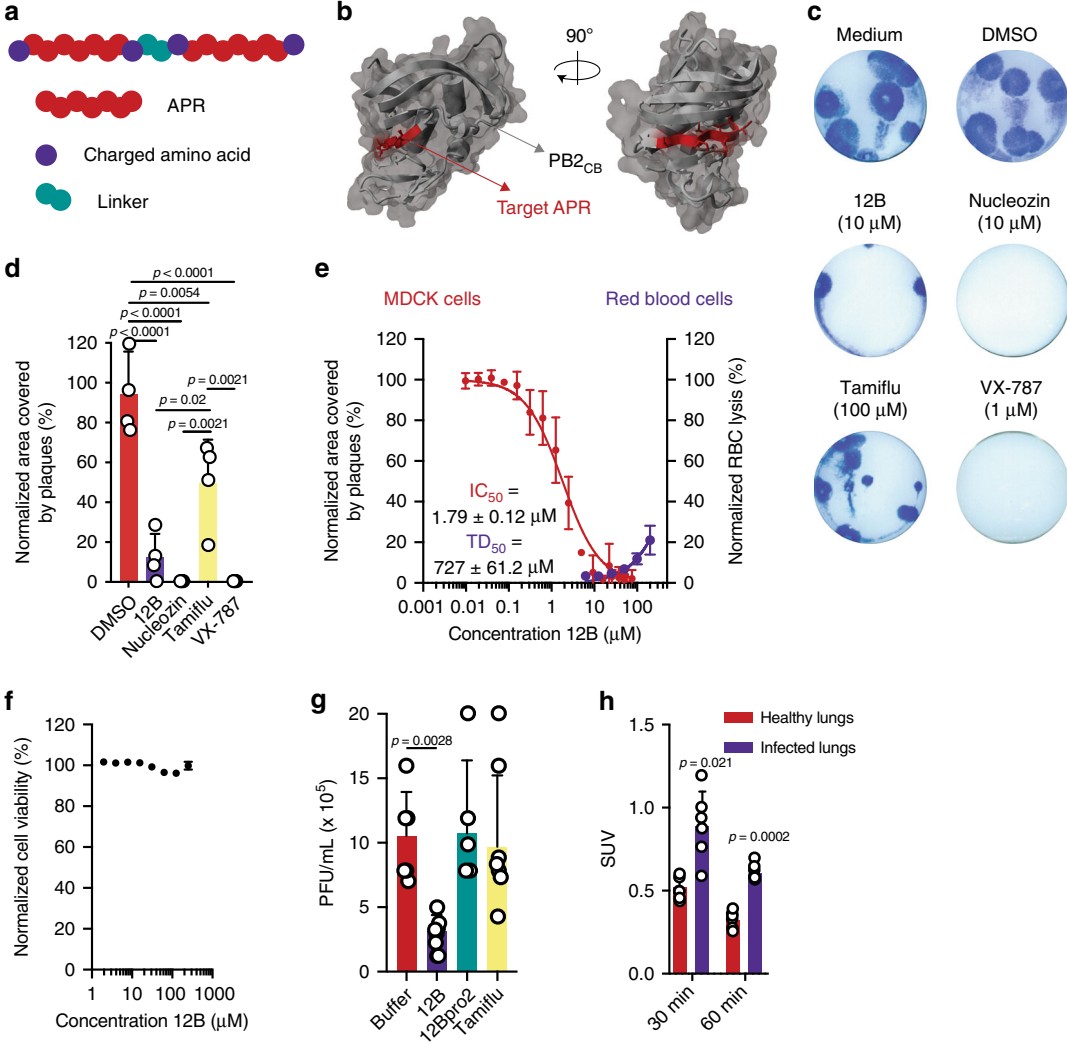

**Fig. 1 APR-based design of a synthetic peptide with antiviral activity against influenza A. a** Graphical illustration of the peptide tandem design with the target APRs capped by charged amino acids (arginine or aspartic acid) and linked by a glycine-serine or proline-proline linker. **b** Cap-binding domain of the influenza A/PR8 polymerase basic protein 2 (PB2$_{CB}$) (PDB = 4ENF)[26,27]. The target APR, $_{381}$LIQLIVS$_{387}$, is surface exposed and highlighted in red. **c** Plaque-size reduction assay of MDCK cells treated with cell medium, 1% DMSO, 10 μM peptide 12B, 10 μM nucleozin, 100 μM Tamiflu, or 1 μM VX-787 2 h prior to infection with influenza A/PR8. **d** Quantification of the area covered by plaques in **c**. Data are normalized to medium-treated cells and mean ± SD is shown (n = 4 independent experiments, statistics: one-way ANOVA with multiple comparison). **e** Dose-dependent effect of peptide 12B on area covered by plaques (red curve, left axis) and on red blood cell (RBC) lysis (purple curve, right axis). For IC$_{50}$: data are normalized to buffer-treated cells and the mean ± SD is shown (n = 4 independent experiments). For toxic dose (TD$_{50}$): data are normalized to buffer-treated (0% lysis) and 0.1% Triton-treated cells (100% lysis) and the mean ± SD is shown (n = 3 independent experiments). **f** Dose-dependent toxicity of peptide 12B, after 24-h incubation on MDCK cells. Data are normalized to buffer-treated (100% viability) and 0.1% Triton-treated cells (0% viability) and the mean ± SD is shown (n = 3 independent experiments). **g** Viral load in the lungs of influenza A/PR8-infected mice after daily injections of either buffer, 2.5 mg/kg peptide 12B, 2.5 mg/kg peptide 12Bpro2 (all i.v.), or 25 mg/kg Tamiflu (oral gavage) for 6 consecutive days. Data are expressed as mean ± SD (n = 13 mice over six independent experiments, statistics: one-way ANOVA with multiple comparison). **h** Relative concentrations (SUV) of peptide [$^{68}$Ga]Ga-NODAGA-PEG$_2$-12B in the lungs of A/PR8-infected versus healthy mice at different time points after peptide injection. The data are expressed as mean ± SD (n = 12 mice over six independent experiments, statistics: two-sided unpaired Student's t test).

evaluation of all major organs revealed no abnormalities (Supplementary Fig. 2a–c and Supplementary Note 1). Biodistribution studies with a radiolabeled variant ([$^{68}$Ga]Ga-NODAGA-PEG$_2$-12B) in influenza A/PR8-infected BALB/c mice demonstrated a lack of accumulation in healthy tissue, fast blood clearance, and a high urinary elimination without kidney retention (Supplementary Fig. 2d, e). The radiolabeled peptide variant also allowed to estimate the half-life of peptide 12B (<10 min in blood), explaining the modest reduction in viral titers observed in vivo (Fig. 1g) and underlines the need for more stable peptide variants

to obtain a more robust therapeutic activity. Moreover, uptake of radiolabeled peptide 12B was approximately twofold larger in influenza A-infected lungs compared with healthy lungs at different time points after peptide injection, consistent with a specific amyloid–virus interaction in vivo (Fig. 1h).

**Peptide 12B organizes into amyloid-like structures**. We next performed biophysical studies of peptide 12B in vitro to verify that it behaves as an amyloid, in accordance with the design. Dynamic light scattering (DLS) showed that peptide 12B (100 μM) organizes

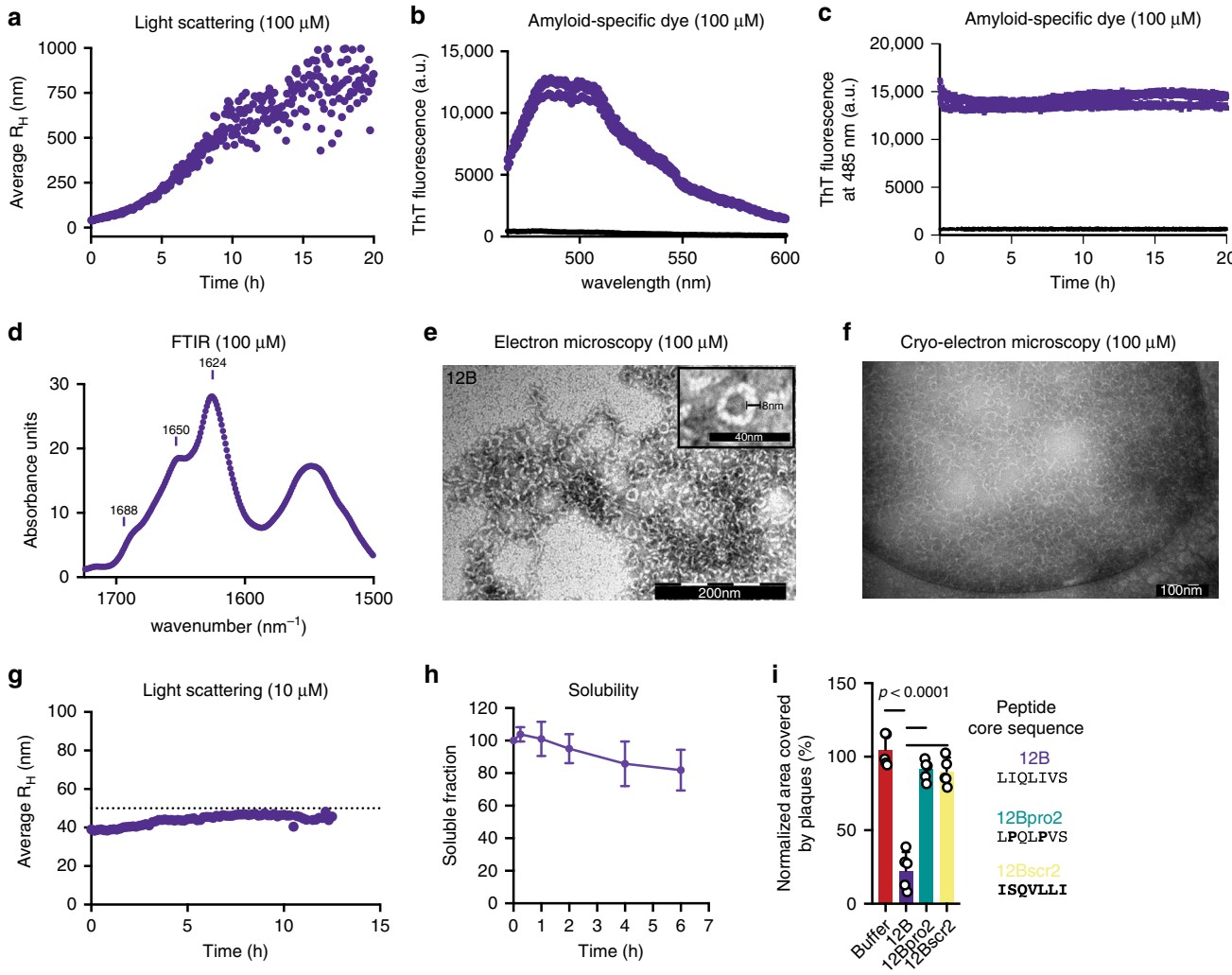

**Fig. 2 Peptide 12B organizes into amyloid-like structures in vitro. a** Hydrodynamic radius ($R_H$) calculated from the regularization fit of DLS data of peptide 12B (100 μM) maturation over time. **b** Th-T emission spectrum after excitation at 440 nm of buffer (black line) or peptide 12B (100 μM, purple line) 5 min after solubilization ($n = 3$ independent experiments). **c** Th-T emission signal of buffer (black line) or peptide 12B (100 μM, purple line) at 485 nm after excitation at 440 nm as a function of time. Data represent three independent repeats ($n = 3$ independent experiments). **d** FTIR spectrum of 100 μM peptide 12B 5 min after solubilization. TEM (**e**) and cryo-TEM (**f**) of 100 μM freshly dissolved peptide 12B (both are one representative image of three independent experiments). **g** Hydrodynamic radius ($R_H$) calculated from the regularization fit of DLS data of peptide 12B (10 μM) maturation over time (dashed line = 50 nm). **h** Soluble fraction of peptide 12B (10 μM) determined after ultracentrifugation (250,000 × $g$ for 30 min), measured over time, and mean ± SD is shown ($n = 3$ independent experiments). **i** Quantification of the area covered by plaques of MDCK cells treated with 10 μM peptide in a plaque-size reduction assay with influenza A/PR8. Data are normalized to medium-treated cells and the mean ± SD is shown ($n = 5$ independent experiments, statistics: one-way ANOVA with multiple comparison).

into small structures (<50 nm) that evolve into larger aggregates (~1 μm) within 20 h (Fig. 2a). Freshly dissolved peptide 12B already binds the amyloid-specific dye thioflavin-T (Th-T) and the Th-T spectra do not change significantly over time (Fig. 2b, c). The Fourier-transform infrared (FTIR) spectrum of a freshly dissolved peptide shows a main peak at 1624 cm$^{-1}$, which is indicative of intermolecular beta-sheet formation and is hence typically observed for amyloid-like aggregates[19] (Fig. 3d). To evaluate peptide aggregate morphology, we used transmission electron microscopy (TEM), showing amyloid-like structures of ±8–10 nm in width (Fig. 3e). However, in contrast to typical amyloid fibers, peptide 12B appears to organize in shorter, curved fibers, sometimes forming annular aggregates[20,21]. Cryo-TEM images confirm that these structures are present in solution and are not an artifact of the drying or staining steps in regular TEM (Fig. 3f). Together, these data indicate that peptide 12B organizes into small beta-sheet-containing structures that grow into larger amyloid-like aggregates

over time. Importantly, DLS results indicate that these small oligomeric structures can be stabilized by simply keeping peptide 12B at a low concentration (10 μM), which is approximately fivefold higher than its IC$_{50}$, as described in the antiviral assays above (Fig. 3g). Finally, these small oligomeric structures are soluble, as shown by ultracentrifugation followed by concentration determination at different time points (Fig. 3h).

**Forming amyloid is necessary but insufficient for antiviral activity.** As controls, we designed two additional peptides, in which we maintained an identical scaffold but altered the APR cores (Supplementary Table 3). Peptide 12Bpro2 was designed as a control for the role of beta-strand-mediated aggregation by introducing two proline mutations in each APR to reduce the aggregation propensity while avoiding the introduction of additional charged residues[22]. Peptide 12Bscr2 was designed as a control for the sequence specificity of aggregation by

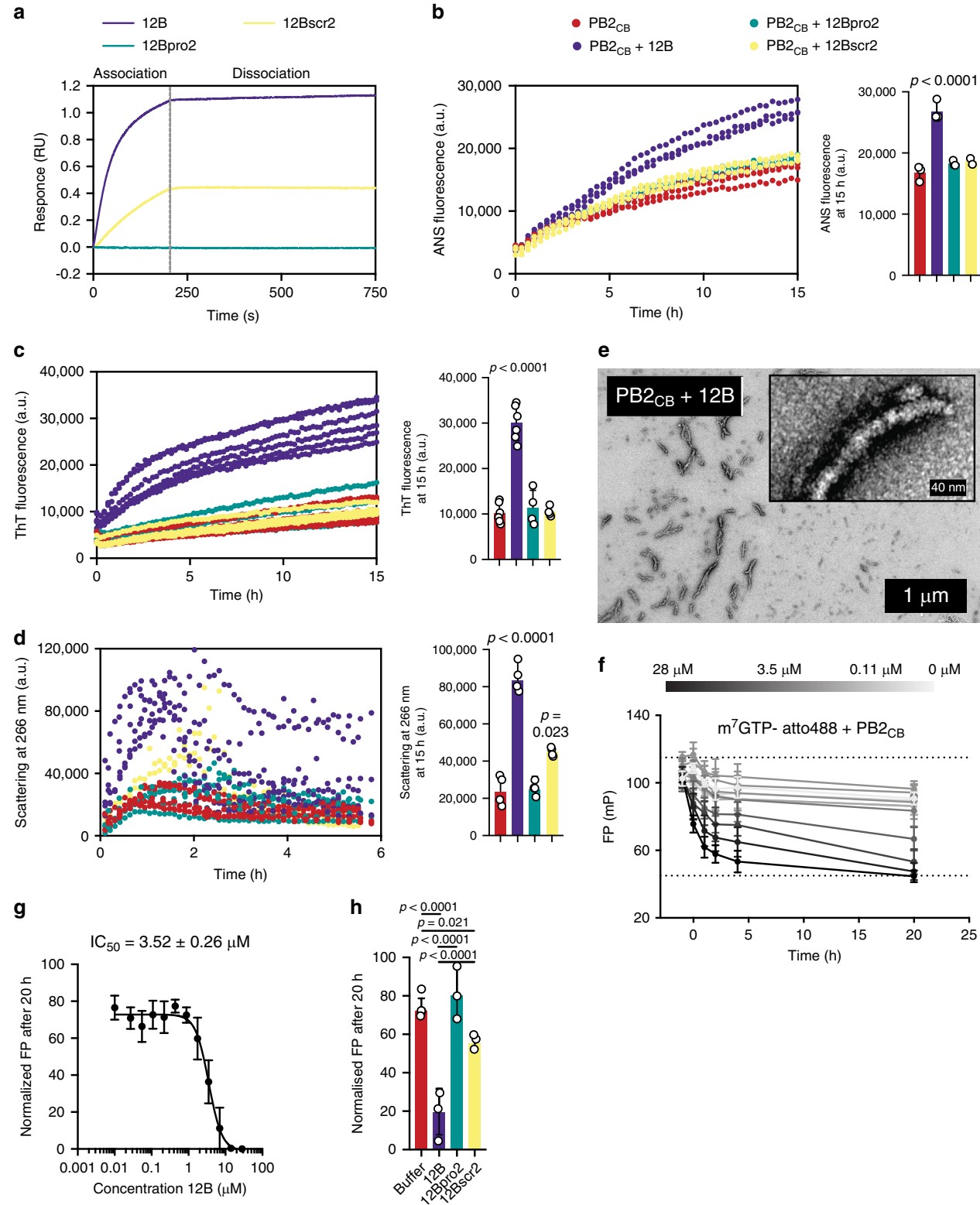

scrambling the amino acids of the APR while maintaining its propensity to form amyloids. As intended by their design, an in vitro biophysical analysis showed that, in our experimental conditions, 12Bscr2 (100 μM) organizes into amyloid-like aggregates over time, while 12Bpro2 (100 μM) does not aggregate (Supplementary Fig. 3a–f). Only at a higher concentration (500 μM), peptide 12Bpro2 organizes into amorphous aggregates (Supplementary Fig. 3d). Treatment of influenza A/ PR8-infected MDCK cells with these control peptides showed no significant effect on viral replication, indicating that peptide amyloid formation is necessary but in itself not sufficient for antiviral activity (Fig. 2i, Supplementary Fig. 3g, h). Moreover, peptide 12Bpro2 did not show inhibitory effects on influenza A replication in the murine infection model (Fig. 1g).

**Fig. 3 Amyloid peptide 12B interacts with influenza via the PB2 target protein in vitro. a** Biolayer interferometry showing association and dissociation of 10 μM peptide to his-tagged, immobilized $PB_{CB}$. **b, c** ANS (excitation 380 nm, emission 485 nm) and Th-T (excitation 440 nm, emission 485 nm) fluorescence of $PB2_{CB}$ (32 μM) with and without peptides (3.2 μM) over time. Quantification of the fluorescence signal after 15 h is shown on the right. Data are expressed as mean ± SD ($n = 3$ (**b**) and $n = 6$ (**c**) independent experiments, statistics: one-way ANOVA with multiple comparison to $PB2_{CB}$ alone). **d** Particle scattering (260 nm) of $PB2_{CB}$ (32 μM) with and without peptides (3.2 μM) over time. Quantification of scattering signal after 1 h (before precipitation) is shown on the right. Data are expressed as mean ± SD ($n = 4$ independent experiments, statistics: one-way ANOVA with multiple comparison to $PB2_{CB}$ alone). **e** TEM images of $PB2_{CB}$ (32 μM) with peptide 12B (3.2 μM) after 15 h incubation. **f** Binding of $m^7GTP$-atto488 by $PB2_{CB}$ in absence or presence (gray scale for peptide concentration) of a twofold serial dilution of peptide 12B, determined by fluorescence polarization (FP). Mean ± SD is shown ($n = 4$ independent experiments). The dashed lines represent maximal binding of $m^7GTP$-atto488 to PB2 (upper dashed line) and free $m^7GTP$-atto488 (lower dashed line), based on control experiments (Supplementary Fig. 4h). **g** Concentration-dependent effect of peptide 12B on the binding of $m^7GTP$-atto488 by $PB2_{CB}$ (5 μM) after 20 h incubation as shown by FP. Data represent normalized values (maximal binding = 100% and free $m^7GTP$-atto488 = 0%) and mean ± SD is shown ($n = 5$ independent experiments). **h** Binding of $m^7GTP$-atto488 by $PB2_{CB}$ (5 μM) with or without peptides (7 μM) after 20 h incubation. Data represent normalized values (similar as in **g**) and mean ± SD is shown ($n = 3$ independent experiments, statistics: one-way ANOVA with multiple comparison).

**Amyloid peptide 12B interacts with influenza via PB2 in vitro.** To provide mechanistic understanding of the amyloid–virus interaction, we used recombinant expression and purification of the PB2 cap-binding domain of influenza A/PR8 ($PB2_{CB}$), the intended target protein of peptide 12B (Supplementary Fig. 4a), and performed direct interaction measurements between the synthetic amyloid 12B and the purified protein. Biolayer interferometry (BLI) showed a clear binding event of amyloid peptides 12B and 12Bscr2 to immobilized $PB2_{CB}$, while 12Bpro2 did not show any affinity for $PB2_{CB}$ (Fig. 3a). The dissociation constant ($K_d$) of peptide 12B to $PB2_{CB}$ (65.8 ± 0.91 nM) is >10-fold lower compared with 12Bscr2 (714 ± 33.9 nM) (Supplementary Fig. 4b, c). $PB2_{CB}$ showed an increased binding of 8-anilinonaphtalene-1-sulfonic acid (ANS) when seeded with amyloid peptide 12B, indicative of a conformational change of the protein that leads to the exposure of hydrophobic residues normally buried in the folded protein (Fig. 3b). In agreement with the latter observation, 12B, and not control peptides 12Bpro2 and 12Bscr2, increased Th-T binding and scattering intensity of $PB2_{CB}$, consistent with beta aggregation (Fig. 3c, d). Importantly, this increase in ANS fluorescence, Th-T fluorescence, and scattering did not originate from the aggregation of the peptides alone (Supplementary Fig. 4d–f). In addition, TEM images showed that the final structures formed by 12B-induced $PB2_{CB}$ aggregates are fiber-like (Fig. 3e), compared with 12Bpro2-, 12Bscr2-, or noninduced $PB2_{CB}$ aggregates, which have an amorphous structure (Supplementary Fig. 4g). All together, these data show that peptide 12B binds with nanomolar affinity to $PB2_{CB}$, inducing a conformational change of the natively folded $PB2_{CB}$, resulting in the formation of beta-aggregates. To assess that this leads to loss of function, we studied the so-called "cap-snatching" process, which occurs at the initiation of viral RNA synthesis, in which PB2 binds host pre-mRNA via a 5′cap structure ($m^7GTP$). To validate whether specific 12B-induced aggregation is associated with a loss of PB2 cap-binding function, we used a fluorescent polarization assay (Supplementary Fig. 4h)[23]. When a fluorescently labeled ligand ($m^7GTP$-atto488) and purified $PB2_{CB}$ were mixed with peptide 12B, a concentration-dependent loss of cap-binding function was observed, which further decreased over time (Fig. 3f, g; Supplementary Fig. 4i). Control peptide 12Bpro2 had no effect on PB2 cap-binding activity, while 12Bscr2 showed only a mild effect, possibly due to the modest affinity of this peptide to $PB2_{CB}$ (Fig. 3h).

**Amyloid peptide 12B interacts with influenza via PB2 in cellulo.** The in vitro results described in the previous section show that amyloid peptide 12B binds to PB2, induces its aggregation, and interferes with its function. Next, we studied whether the same event occurs in the cellular cytoplasm. To establish that in

a cellular context the synthetic amyloid does not interfere with viral entry in an aspecific manner, we performed a time of addition assay. This assay shows that the antiviral activity of 12B is unchanged, whether it is used in a pretreatment setup or added 6 h after infection (Fig. 4a, Supplementary Fig. 5a). Moreover, when influenza A virion particles are pretreated with amyloid peptide 12B, the antiviral activity does not change dramatically compared with pretreatment of the cells (Fig. 4b, Supplementary Figs. 1e and 5b). All the later indicates that 12B does not aspecifically interfere with viral entry and suggests that the amyloid–virus interaction occurs in the cell. To study the interactions between amyloid peptide 12B and influenza A in the cell and to confirm that this interaction also induces aggregation and inactivation of PB2, we used a modified influenza A/WSN/1933(H1N1) strain, with a FLAG-tagged PB2 (A/WSN-FLAG), facilitating PB2 detection. Immunofluorescence experiments showed co-localization of a FITC-labeled peptide 12B variant with PB2 (Fig. 4c). To support the latter observation, we used a $PEG_2$-biotin labeled variant of peptide 12B, 12Bpro2, and 12Bscr2 to perform a co-immunoprecipitation of PB2 from lysates of peptide-treated, A/WSN-FLAG-infected MDCK cells. The enrichment of PB2 in the co-immunoprecipitated fraction of 12B-$PEG_2$-biotin-treated MDCK cells is consistent with a direct interaction between amyloid peptide 12B and the target protein PB2 (Fig. 4d). Finally, peptide 12B treatment of A/WSN-FLAG-infected MDCK cells resulted in a reduction of soluble PB2, while the control peptides had no effect (Fig. 4e), showing PB2 undergoes aggregation. Taken together with the significant reduction in viral replication upon treatment with 12B shown above (Fig. 1c, d), these data confirm that amyloid peptide 12B interacts with PB2 in the cytoplasm of influenza A-infected MDCK cells and induces its inactivation through aggregation.

**Amyloid peptide 12B acts in an APR-specific manner.** According to our model, the underlying properties of amyloid interactions confer specificity upon the amyloid–virus interaction, but it was unclear how much tolerance to sequence variation is present. An in silico analysis of over 40,000 influenza A strains showed that the target APR in PB2 is highly conserved (Supplementary Fig. 6a). We evaluated the antiviral activity of amyloid peptide 12B on a set of influenza A strains, across different subtypes (H1N1, H3N2, H3N3, H3N8, H5N1), in which the frequently occurring single polymorphisms of the target APR are represented ($_{381}LIQLIVS_{387}$ to $_{381}LVQLIVS_{387}$ or $_{381}LFQLIVS_{387}$). Peptide 12B retains activity against all these influenza A strains, suggesting the interaction allows for some tolerance (Fig. 5a). In addition, we performed an in silico mutational analysis in which the effect of two mutations in the APR on the interaction potential

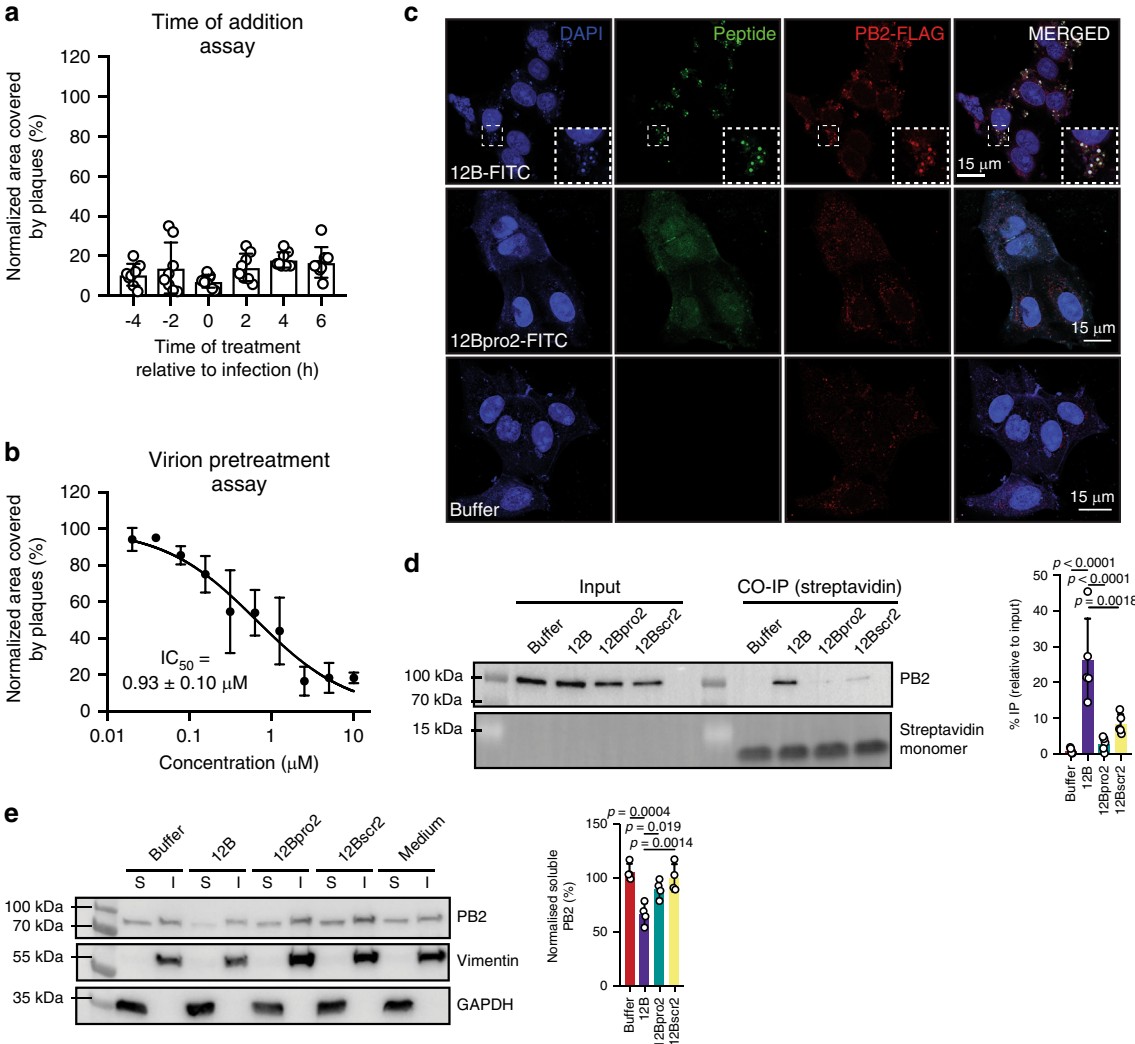

**Fig. 4 Amyloid peptide 12B interacts with influenza via the PB2 target protein in cellulo. a** Quantification of the area covered by plaques of A/PR8-infected MDCK cells treated with 10 μM peptide 12B at different time points relative to infection. Data are normalized to medium-treated cells and mean ± SD is shown ($n = 8$ from three independent experiments, statistics: one-way ANOVA with multiple comparison, $p$ value = 0.0605). Representative images are shown in Supplementary Fig. 5a. **b** Quantification of the area covered by plaques in a plaque-size reduction assay of MDCK cells infected with A/PR8 virion particles that were pretreated with peptide 12B. Concentration-dependent effect of peptide 12B is shown normalized to medium-treated virion particles as mean ± SD ($n = 3$ independent experiments). Representative images are shown in Supplementary Fig. 5b. **c** MDCK cells treated with FITC-labeled peptide 12B, 12Bpro2 (10 μM), or buffer for 2 h prior to A/WSN-FLAG infection (MOI = 1, 16-h infection). Cells were fixed and stained with an anti-FLAG antibody and DAPI. One representative example of three independent experiments. **d** Western blot analysis of total (input) and immunoprecipitated fraction of MDCK cells treated with PEG$_2$-biotin-labeled peptide (10 μM) or buffer for 2 h prior to A/WSN-FLAG infection (MOI = 1, 16-h infection). Quantification of immunoprecipitated fraction of PB2, relative to PB2 input levels, is shown on the right. Data represent mean ± SD ($n = 5$ independent experiments, statistics: one-way ANOVA with multiple comparison). Streptavidin was used as a loading control. **e** Western blot analysis of PB2 distribution in soluble and insoluble fraction of lysates of MDCK cells treated with peptide (10 μM) for 2 h prior to influenza A/WSN-FLAG infection (MOI = 1, 16-h infection). Vimentin and GAPDH were used as loading controls for insoluble and soluble fraction, respectively. Quantification represents soluble PB2 fraction, normalized to medium-treated cells (100% soluble PB2). Data represent mean ± SD ($n = 4$ independent experiments, statistics: one-way ANOVA with multiple comparison). For **d** and **e** the samples on one blot are derived from the same experiment and the gels/blots were processed in parallel. Full blots are shown in Supplementary Fig. 12.

was calculated (Supplementary Fig. 7, Supplementary Note 2). Only 17.3% of all possible double-mutant combinations are predicted to still interact with the $_{381}$LIQLIVS$_{387}$ sequence, of which >90% would have a strong negative effect on PB2 protein stability, and hence do not occur in natural influenza A strains. On the other hand, 12B did not interfere with the replication of several influenza B virus strains (B/Memphis/12/97 (B/Mem), B/Wisconsin/01/2010 (B/Wis), and B/Brisbane/60/2008 (B/Bris)), in which all have ~40% overall sequence homology to influenza A/PR8 but lack the specific target APR, in vitro (Fig. 5a, b). Of note,

the corresponding target sequence in influenza B ($_{383}$MEKLLIN$_{389}$) is located in the same position in the cap-binding domain of PB2 compared with the target sequence ($_{381}$LIQLIVS$_{387}$) in influenza A PB2. In addition to the plaque-size reduction assay, influenza B replication is also unaffected in a multicycle replication assay and 12B shows no protection against cytopathic effects of influenza B (Supplementary Fig. 6b, c). Of note, time of addition assays and pretreatment of influenza B virion particles showed no antiviral activity of amyloid peptide 12B, again confirming that our amyloid does not act as a nonspecific inhibitor of

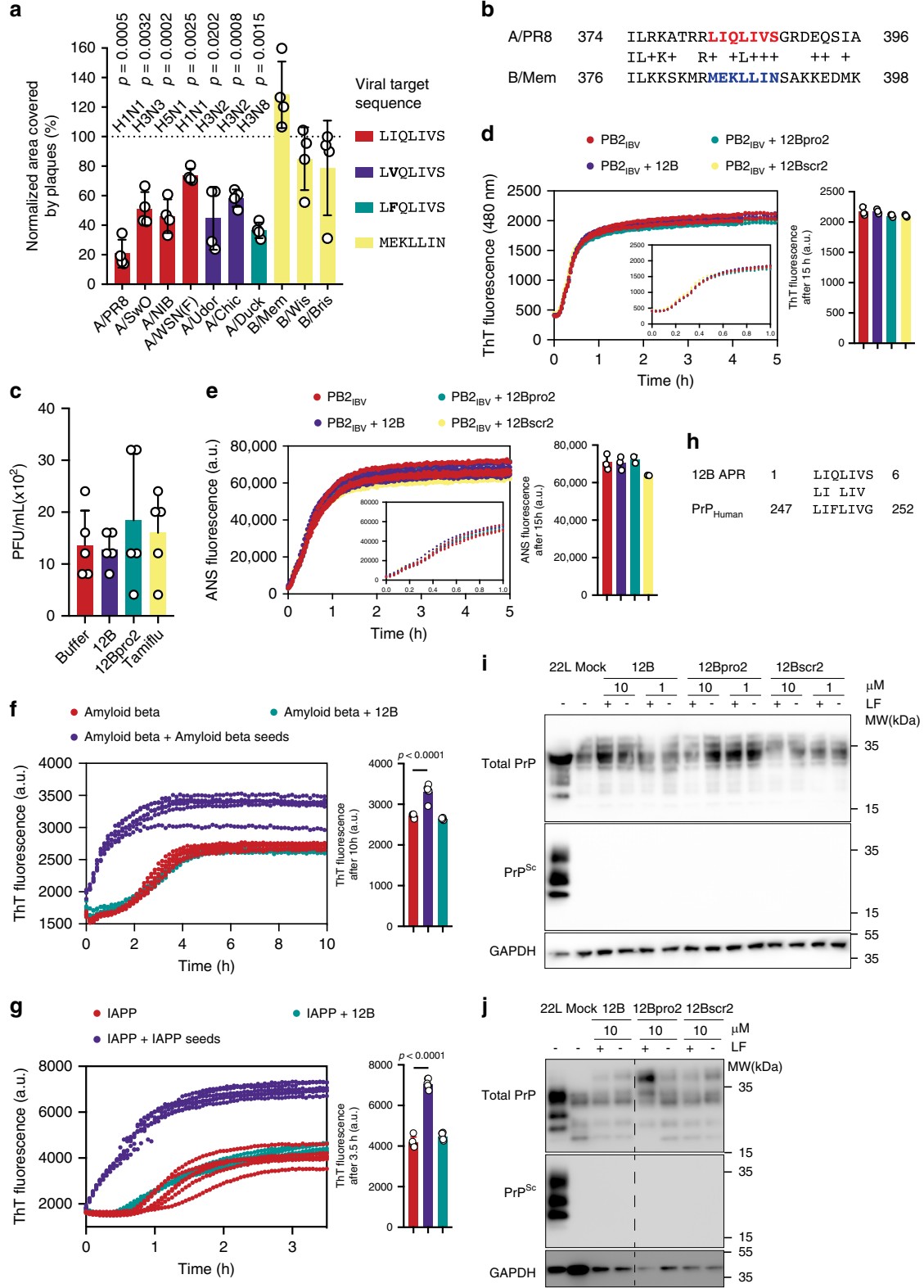

viral entry (Supplementary Fig. 5a, b). Influenza B replication is also not affected by peptide 12B in the murine infection model, confirming the APR-specific activity in vivo (Fig. 5c). As expected from the lack of target APR in the protein, peptide 12B showed no effect on the aggregation or conformational stability of recombinantly purified PB2$_{CB}$ from influenza B/Mem (PB2$_{IBV}$) as seen from Th-T and ANS binding assays (Fig. 5d, e). In the same line, peptide 12B was not able to affect the aggregation of the highly

aggregation-prone amyloids when they share no similarity with peptide 12B (Aβ$_{42}$ or amylin (IAPP), Fig. 5f, g, respectively). To explore the limits of the sequence-identity tolerance, we searched for a naturally occurring aggregation-prone protein with imperfect identity to peptide 12B. The C-terminal domain (glycosylphosphatidylinositol-anchor sequence) of the prion protein (PrP) contains a region with high sequence homology to peptide 12B, both in human and mouse, covering only two mismatches (Q249F

**Fig. 5 Amyloid peptide 12B acts as a sequence-specific interferor of influenza A. a** Quantification of area covered by plaques of influenza-infected MDCK cells, treated with 10 μM peptide 12B. Data are normalized to buffer-treated cells and mean ± SD is shown ($n = 4$ independent experiments, statistics: two-sided one-sample $t$-test). A list of abbreviations is provided in "Methods." **b** Local alignment of PB2 from influenza A/PR8 and B/Mem. Target APRs are highlighted in red (A/PR8) and blue (B/Mem). **c** Viral load in the lungs of influenza B/Mem-infected mice after daily injections of buffer, 2.5 mg/kg peptide 12B, 2.5 mg/kg peptide 12Bpro2 (i.v.); or 25 mg/kg Tamiflu (oral gavage) for 6 days. Data represent mean ± SD ($n = 13$ mice, five independent experiments, statistics: one-way ANOVA with multiple comparison, $p$ value = 0.6027). **d**, **e** Th-T and ANS fluorescence of PB2$_{CB}$(IBV) (32 μM) without and with peptides (3.2 μM, enlarged in the inserts). Fluorescence after 20 h is shown on the right as mean ± SD ($n = 3$ independent experiments, statistics: one-way ANOVA with multiple comparison, $p$ value = 0.4218 for **d** and 0.1499 for **e**). **f**, **g** Th-T fluorescence of amyloid beta (10 μM) and IAPP (10 μM) without and with peptide 12B (1 μM) is shown. Self-seeded control is included by using preformed aggregates of amyloid beta and IAPP, respectively (1 μM). Th-T fluorescence at plateau is shown on the right as mean ± SD ($n = 4$ independent experiments, statistics: one-way ANOVA with multiple comparison). **h** Local alignment of the target APR in peptide 12B with human prion protein (PrP$_{human}$). Western blot analysis of proteinase K-treated lysates of L929 15.9 cells (six passages, (**i**)) and CAD5 cells (eight passages, (**j**)) post treatment with peptides (1 or 10 μM) with or without Lipofectamine 2000 (LF). As controls, cells were exposed to brain homogenate from a terminally diseased mouse infected with scrapie strain 22L (PrP$^{Sc}$) or with normal (Mock) brain homogenate. GAPDH was used as loading control. The samples are derived from the same experiment and the gels/blots were processed in parallel. Full blots are shown in Supplementary Fig. 12.

and S253G, Fig. 5h). However, peptide 12B was incapable to convert PrP to the aggregated prion state (PrP$^{SC}$) in cells expressing soluble mouse PrP$^C$ (Fig. 5i, j). The combination of a sequence-specific process (not affecting influenza B or PrP) that is tolerant to single conservative substitutions (affecting the influenza A variants) suggests that the approach has potential as a broad-spectrum antiviral strategy to target influenza A.

**Exploring the chemical space of antiviral synthetic amyloids**. In order to get an indication if our approach with Influenza A is generally transferable, we set out to identify amyloids that target the ZIKV specifically. Following the same approach as described earlier, 63 different peptides were designed based on APRs identified in the ZIKV proteome (African strain MR766, Supplementary Table 4). Vero E6 cells were treated with peptide at a concentration of 20 μM, 2 h prior to infection. Following infection for 48 h, the number of infected cells was determined by immunofluorescence. Multiple peptides showed inhibitory effects on ZIKV replication and protective effects toward the infected mammalian cell line (Supplementary Figs. 8 and 9). Overall, peptide R50 (RVAIAWLLRGSRVAIAWLLR), derived from an APR identified in membrane glycoprotein M, $_{140}$VAIAWLL$_{146}$, showed the most pronounced effect compared with our positive control compound 7-DMA (7-deaza-2′-C-methyladenosine)[24]. A biophysical analysis confirmed the amyloid propensity of peptide R50. DLS showed that peptide R50 organizes into small structures (<50 nm) that evolve into larger aggregates (~300 μm) within 5 h (Fig. 6a). These aggregates are positive for the amyloid-specific dye Th-T (Fig. 6b) and have an amyloid-like morphology as seen from TEM images (Fig. 6c). Even though peptide R50 organizes into amyloid structures, ~80% of total peptide remains in solution over time (Fig. 6d). A dose–response experiment, in which Vero E6 cells were treated with different concentrations of peptide or compound, prior to viral infection showed IC$_{50}$ values of 5.25 ± 0.30 and 4.74 ± 0.31 μM for peptide R50 and 7-DMA, respectively (Fig. 6e, f, Supplementary Fig. 10a, b). Cross-validation with peptide 12B confirmed the APR-specificity associated with our approach as no interference with ZIKV replication is observed with this anti-influenza peptide (Fig. 6g, Supplementary Fig. 10c). Importantly, the concentration-dependent inhibitory effect of amyloid peptide R50 and 7-DMA on ZIKV replication did not originate from aspecific toxicity on the mammalian host cell (Fig. 6h–j). Finally, reminiscent of amyloid peptide 12B, a time of addition assay, shows that R50 remains active, even when added 6 h after ZIKV infection (Supplementary Fig. 11a, b).

From these data it appears that our design approach is generally applicable and can be used to specifically target different viruses. In other words, these family of antiviral peptides constitutes a novel chemical space that could be mined for novel antivirals that act through the mechanism of targeted protein aggregation.

## Discussion

Human encoded amyloids possess functional roles and are not just pathogenic byproducts in disease[1]. The AD-associated amyloid, Aβ, physically interacts with herpes virus particles and interferes with viral replication. Building on this observation, we present here the reverse engineering of synthetic amyloids that specifically interact with a predefined target virus and reduce viral replication. Since it is well established that amyloid interactions are driven by the sequence-specific assembly of homologous aggregation-prone fragments[8–12], we designed our amyloids by encoding a unique amino acid fragment corresponding to a viral APR. Using influenza A as a model virus, we designed an amyloid with a homologous APR to the PB2 protein. Our results show that this homologous APR drives the interaction between the amyloid and PB2, leading to aggregation and loss-of-function of the PB2 protein. We show that our amyloid does not aspecifically interfere with virus entry, instead amyloid–virus interaction occurs in the cytosol of the infected cell. Following amyloid uptake[25], a sequence-specific binding event induces PB2 aggregation in the cytosol. Since APRs are usually buried inside the core of the folded protein, we suspect that this interaction happens during protein translation[15], when the APR might still be solvent exposed. PB2 loses its essential cap-snatching function following the amyloid-induced aggregation, leading to reduced viral replication. Importantly, our amyloid interacts with PB2 in a sequence-specific manner, since two mutations are sufficient to abrogate the interaction. Finally, the universal mechanism underlying homotypic APR interactions allowed us to extrapolate this technology resulting in a second amyloid that inhibits ZIKV replication.

Aβ encodes an APR matching an APR in envelope glycoprotein B, a herpes protein[4]. Our results indicate that such homolog APRs are sufficient to drive the amyloid–virus interaction and provide a mechanistic understanding how such interactions would arise. Whether the suggested mechanism of interaction is broadly used for amyloid–virus interactions like the Aβ amyloid and the herpes glycoprotein B remains to be determined together with the general role of amyloids in innate immunity. However, our work demonstrates that the increasing understanding of the functional aspects of amyloids could extend beyond purely amyloid-specific structural roles such as scaffolding or conformational switches[1,26–28]. Instead, amyloid structures also entail the ability to engage in very specific interactions with other proteins under complex physiological conditions and affect their

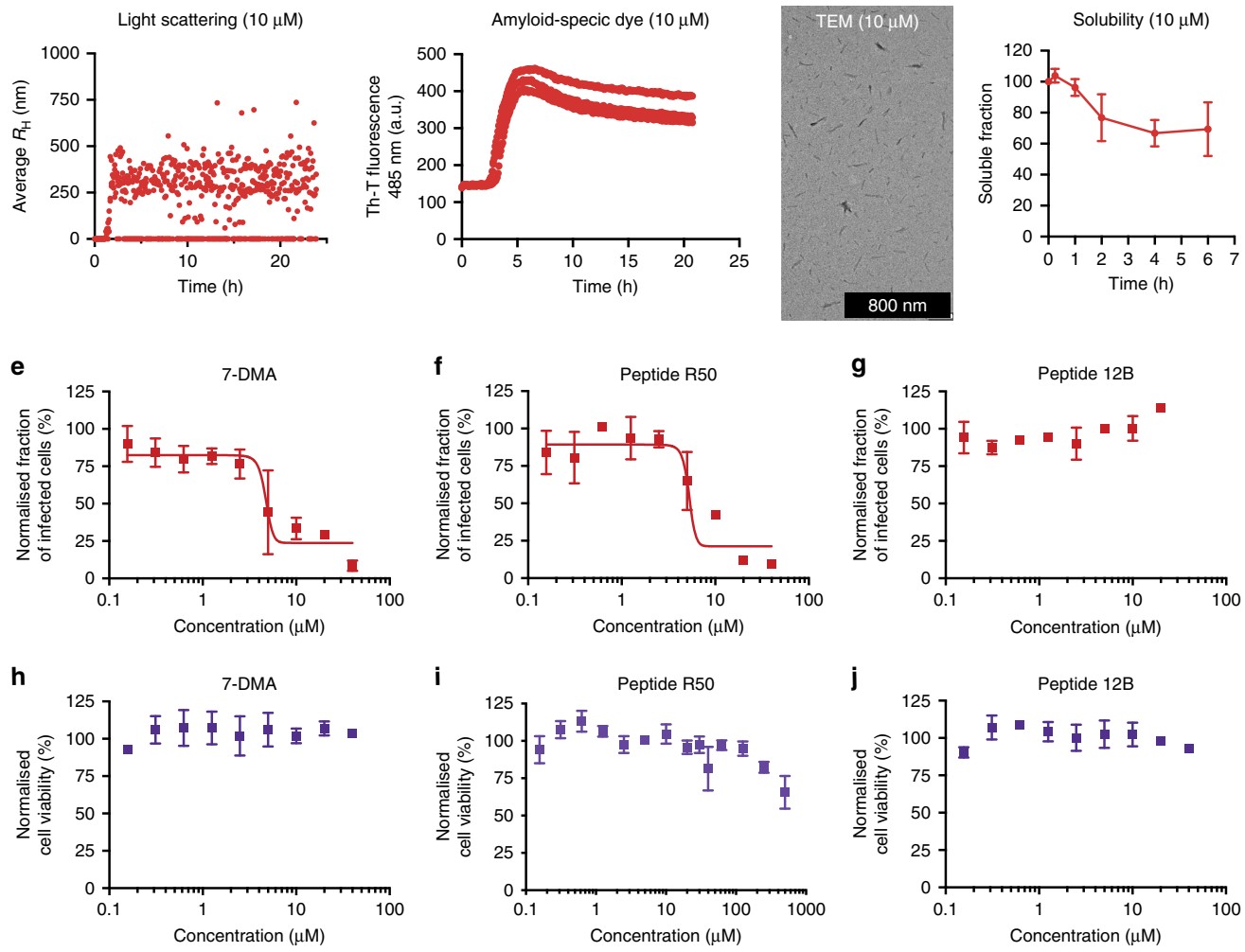

**Fig. 6 Peptide R50 organizes into amyloid structures and interferes with ZIKV replication. a** Hydrodynamic radius ($R_H$) calculated from the regularization fit of DLS data of peptide R50 (10 μM) maturation over time. **b** Th-T emission signal of 10 μM peptide R50 at 485 nm after excitation at 440 nm as a function of time ($n = 3$ independent experiments). **c** TEM image of 10 μM peptide R50 after 6 h incubation. **d** Soluble fraction of peptide R50 (10 μM) determined after ultracentrifugation (250,000 × $g$ for 30 min), measured over time, and mean ± SD is shown ($n = 3$ independent experiments). Dose-dependent effect of 7-DMA (**e**), peptide R50 (**f**), and peptide 12B (**g**) on the fraction of ZIKV-infected cells following 48 h of infection with a MOI of 0.1. Data are normalized to DMSO-treated, infected cells (100% infected cells) and mean values ± SD are shown ($n = 3$ independent experiments). Representative images are shown in Supplementary Fig. 10. Dose-dependent effect of 7-DMA (**h**), peptide R50 (**i**), and peptide 12B (**j**) on cell viability of Vero E6 cells without ZIKV infection. Data are normalized to DMSO-treated, noninfected cells (100% cell viability) and the mean values ± SD are shown ($n = 3$ independent experiments).

biological function. Finally, the ability to reverse engineer synthetic amyloids with specific biological activity also offers a new toolkit for protein functional interference and possibly therapeutic intervention.

## Methods

**Reagents**. All reagents and antibodies are listed in Supplementary Table 5.

**Mouse models**. All mouse experiments were conducted according to the National (Belgian Law 14/08/1986 and 22/12/2003, Belgian Royal Decree 06/04/2010) and European (EU Directives 2010/63/EU, 86/609/EEG) animal regulations. All protocols were approved by the Institutional Ethics Committee of Ghent University (Eth. Com. no. 2018-010). Five to eight weeks old healthy female wild-type BALB/C mice were used in all mice experiments. Three or four mice were housed together in standardized cages with unrestricted access to food and water.

**Tissue culture**. MDCK cells were obtained from the American Type Culture Collection (ATCC). MDCK cells were maintained at 37 °C with 5% CO2 in DMEM medium (GIBCO), supplemented with 10% FCS (GIBCO), L-glutamine (GIBCO), and nonessential amino acids (GIBCO). HEK 293T cells were obtained from the

ATCC and maintained in DMEM medium, supplemented with 10% FBS, 1 mM sodium pyruvate, and nonessential amino acids. Mouse fibroblast cell line L929 (ECACC; L929 (NCTC); #85103115) was purchased from Sigma-Aldrich (Taufkirchen, Germany). The L929 subclone 15.9, which is highly susceptible to the mouse-adapted scrapie strain 22L, was used for all experiments[29,30]. CAD5 cells[31] were a kind gift of Corinne Lasmezas (The Scripps Research Institute, FL).

**Bioinformatics**. The statistical thermodynamics algorithm, TANGO, was used for all APR identifications. A cutoff on the TANGO score of five per residue was selected, which results in a Matthews correlation coefficient between prediction and experiment of 0.92. Settings within TANGO: temperature = 298 K; pH = 7.5; ionic strength = 0.15 M; and minimum length of APRs = 5.

**Viruses**. Viruses used throughout the work include A/Puerto Rico/8/34 (H1N1): A/PR8, A/NIBRG-14 (H5N1): A/NIB, A/Swine Ontario/42729A/2001 (H3N3): A/SwO, A/Udorn/307/72 (H3N2): A/Udor, A/Chicken/Nanchang/3-120/2001 (H3N2): A/Chic, A/Duck/Ukraine/1/63 (H3N8): A/Duck, A/WSN-FLAG (H1N1): A/WSN(F), B/Memphis/12/97 (Yamagata-lineage): B/Mem, B/Wisconsin/01/2010 (Yamagata-lineage): B/Wis, and B/Brisbane/60/2008 (Victoria-lineage): B/Bris. A/NIBRG is an influenza stain derived from a highly pathogenic H5N1 strain (A/Vietnam/1194/2004), in which the HA is adapted to a low-pathogenic form. Only

HA and NA are derived from the pathogenic strain, the other genes are derived from A/PR8. A/WSN-Flag, a recombinant version of A/WSN/1933 (H1N1) carrying a Flag-tagged PB2 protein, was a kind gift of Nadia Naffakh. All viruses were propagated on MDCK cells. All influenza virus stocks were grown in specific-pathogen free eggs, harvested and purified.

**Cell viability assay**. All peptide treatments were done in DMEM medium without serum. Toxicity of the peptide treatments was evaluated using the CellTiter-Blue Cell Viability Assay according to the instructions of the manufacturer (Promega, USA). Briefly, cells were seeded to ~20,000 cells per well in a 96-well plate and allowed to attach overnight. Peptides were diluted in cell medium without serum (concentration indicated in the corresponding figure) and cells were treated for ~24 h, unless stated otherwise. Twenty microliters of the CellTiter-Blue reagent was added to each well and the plate was incubated for 1 h. Finally, fluorescence was measured at 590 nm by exciting at 560 nm with a ClarioStar plate reader (BMG Labtech, Germany).

**Plaque-size reduction assay**. MDCK cells were seeded in complete DMEM medium in 24-well format (160,000 cells per well) and allowed to attach overnight. Cells were washed once with serum-free DMEM medium, treated with peptide (10 μM, unless stated otherwise), and infected with virus (25 pfu, unless stated otherwise) 2 h later. Two hours after infection, all the remaining peptide and virus were washed away with DMEM medium and an overlay of 0.6% avicel[32] in serum-free DMEM medium supplemented with 2 μg/mL TPCK-treated trypsin was added. Virus was allowed to replicate and 48–72 h later cells were fixed and stained for ribonucleoprotein (RNP) using goat anti-RNP (BEI Resources, NR-3133) for influenza A virus or convalescent murine serum for influenza B virus. After staining with anti-goat-HRP (Santa Cruz Biotechnology) or anti-mouse-HRP (GE Healthcare), plaques were visualized using True-Blue HRP substrate (SeraCare). Quantification of the area covered by the plaques was performed by the "colony area" plug-in in ImageJ.

**Hemolytic assay**. Fresh blood was pooled from healthy volunteers (Red Cross Flanders) and erythrocytes were collected by centrifugation $1000 \times g$ for 5 min. EDTA was added as an anticoagulant. The cell pellet was washed three times with phosphate-buffered saline (PBS) and diluted to a concentration of 8% in PBS. One hundred microliters of 8% red blood cells solution was mixed with 100 μL of serial dilutions of peptides in PBS buffer in 96-well plates (BD Biosciences). The reaction mixtures were incubated for 1 h at 37 °C, centrifuged for 10 min at $1000 \times g$, and 100 μL of supernatant was transferred to a sterilized 96-well plate (BD Biosciences, flat bottom). The release of hemoglobin was determined by measuring the absorbance of the supernatant at 495 nm. Erythrocytes in 1% Triton and maximum used concentration of buffer were used as the control of 100% and 0% hemolysis, respectively.

**Biophysical characterization**. A DynaPro DLS plate reader instrument (Wyatt, Santa Barbara, CA, USA) equipped with an 830 nm laser source was used to determine the hydrodynamic radius ($R_H$) of the peptide particles. Two hundred microliters of each sample (at 100 or 10 μM, unless stated otherwise) was placed into a flat-bottom 96-well microclear plate (Greiner, Frickenhausen, Germany). The autocorrelation of scattered light intensity at a 32° angle was recorded for 5 s and averaged over 20 recordings to obtain a single data point. The Wyatt Dynamics v7.1 software was used to calculate the hydrodynamic radius by assuming linear particles. The amyloid-specific dye Th-T was used to study the aggregation state of peptides. Two hundred microliters of each peptide sample (at 100, unless stated otherwise) was placed into a flat-bottom 96-well microclear plate (Greiner, Frickenhausen, Germany) and the dye was added to a final concentration of 25 μM. A ClarioStar plate reader (BMG Labtech, Germany) was used to measure fluorescence by exciting the samples at 440–10 nm and fluorescence emission was observed at 480–10 nm (or a complete spectrum ranging from 470–600 nm). Aggregation kinetics were obtained by placing 200 μL of the peptide solution with a final concentration of 25 μM Th-T into a flat-bottom 96-well microclear plate. Fluorescence emission was monitored at 480–10 nm after excitation at 440–10 nm. Every 5 min Th-T fluorescence was measured. FTIR spectroscopy was performed using a Bruker Tensor 27 infrared spectrophotometer (Bruker, Germany) equipped with a Bio-ATR II accessory (Harrick Scientific Products, USA). Spectra were recorded in the range of 1000–3500 cm$^{-1}$ at a spectral resolution of 2 cm$^{-1}$ by accumulating 256 data acquisitions. The spectrophotometer was continuously purged with dried air. Spectra were corrected for atmospheric water vapor interference, baseline-subtracted, and vector normalized in the amide II area (1500–1600 cm$^{-1}$) as implemented in OPUS software (Bruker). All measurements were performed with 30 μL peptide (at 100, unless stated otherwise) and at room temperature. Solubility of the peptides was quantified by determining the supernatant concentration following ultracentrifugation ($250,000 \times g$ for 30 min).

**Transmission electron microscopy**. Formvar film coated 400-mesh copper grids (Agar Scientific Ltd., England) were first glow-discharged to improve adsorption efficiency. Next, 10 μL of each sample was adsorbed for 5 min and afterward the grids were washed by contact with three drops of ultrapure water. Finally, negative

staining was performed by contact with one drop of uranyl acetate (2% w/v) for 1 min. The grids were examined using a JEM-1400 transmission electron microscope (JEOL, Japan) at accelerating voltage 80 keV.

**Cryo-electron microscopy**. 3.5 μL of sample was applied to a 300-mesh quantifoil grid and incubated for 30 s. Next, grids were mounted in a plunge freezer (EMBL), blotted one-sided for 3 s using a Whatman 1 filter paper, and plunged into liquid ethane at a temperature of −180 °C, and stored in liquid nitrogen until observation in the electron microscope. The samples were transferred to a Gatan 914 cryo-holder and imaged at low-dose conditions at −177 °C, using a JEOL JEM-1400 TEM equipped with an 11 Mpxl Olympus SIS Quemesa camera.

**Aβ 42 and amylin preparation**. The lyophilized Aβ42, Aβ42-HiLyte647, and amylin peptide were purchased from rPeptide (A-1163-1), AnaSpec (AS-64161), and EuroGenTec (AS-60804), respectively, and its suspension and monomeric sample isolation was preformed accordingly with a standard procedure previously described by our group[33]. Briefly, each vial content was suspended in 1,1,1,3,3,3-hexafluoro-2-propanol (HFIP), evaporated under a nitrogen stream, and solubilized in DMSO to a final concentration of 1 mg/mL. DMSO was removed using a 5-mL HiTrap desalting column (GE Healthcare), where the peptide was eluted in 10 mM Tris (pH 7.5) buffer. The concentration of the final monomeric sample was quantified by nanodrop (at 280 nm). The peptide was immediately diluted to 25 μM using the same Tris buffer, and further diluted in the required concentration for the next assays.

**Binding studies**. Binding of peptides to recombinant PB2$_{CB}$ was determined with BLI by using an Octet RED96 instrument (ForteBio, Pall Life Sciences) at 25 °C. His-tagged PB2$_{CB}$ was captured on a nickel-coated (Ni-NTA) biosensor (ForteBio, Pall Life Sciences) in a 50 mM Tris buffer (pH 8.0 and 200 mM NaCl) for 300 s. After three washing steps in peptide buffer (50 mM Tris pH 8.0; 20 mM guanidine thiocyanate), association and dissociation of the peptides to his-tagged PB2$_{CB}$ was measured for 200 and 550 s, respectively. Biosensors without PB2$_{CB}$ were used to control for aspecific binding of peptides to the sensor (data shown are corrected for aspecific binding). Dissociation constant ($K_d$) values were calculated using the Data Analysis software (Octet Software 9.0) by fitting the data to a 1:1 binding model.

**Seeding assay**. All seeding experiments were performed by mixing freshly purified PB2 protein (32 μM, unless stated otherwise) with 10% molar ratio freshly dissolved peptide. For ANS measurements, 25 μM of the dye was added to this mixture and fluorescence was measured by exciting at 380–10 nm and monitoring emission at 480–10 nm. For Th-T measurements, 25 μM of the dye was added to the peptide–protein mixture and fluorescence was recorded by exciting at 440 nm and monitoring emission at 480 nm. Light scattering at 266 nm was monitored with the Optim1000 (Unchained Labs) by using a 266 nm laser.

**Cap-binding assay**. Cap-binding activity of PB2$_{CB}$ was checked by incubation of 20 nM m$^7$GTP-Atto488 with different concentration of purified PB2$_{CB}$-IAV. Fluorescent polarization of the substrate (m$^7$GTP-Atto488) was measured with a PolarStar Optima plate reader using 485–10 nm excitation and 520–10 nm emission filters (BMG Labtech, Germany). Different concentrations of peptides were added to this m$^7$GTP-Atto488-PB2$_{CB}$ mixture (5 μM of PB2$_{CB}$) and the effect on polarization was measured. These data points were always compared with the effect of peptide alone on polarization of m$^7$GTP-Atto488.

**PB2 solubility**. MDCK cells were plated at a density of 300,000 cells/well in a six-well plate 1 day before the experiment. Cells were treated with 10 μM peptide and infected with virus (A/WSN-FLAG, MOI = 1). After 16 h, cells were lysed with NP-40 buffer, supplemented with Complete protease inhibitor (Roche), for 30 min on ice. Cell lysates were centrifuged ($20,000 \times g$) for 20 min at 4 °C and the supernatant was isolated as "soluble fraction." The pellet was washed 3× with PBS and finally dissolved in 30 μL 8 M urea for 1 h ("insoluble fraction"). All samples were heated to 95 °C in SDS-PAGE loading buffer (2% SDS) for 10 min. The samples were separated by SDS-PAGE and anti-FLAG antibody (anti-FLAG, Cell Signaling D6W5, 1:1000) was used to detect PB2. All blots were also stained with an anti-vimentin antibody (anti-Vimentin, Santa Cruz V9, 1:5000), as a control for insoluble fraction, and an anti-GAPDH antibody (anti-GAPDH, Santa Cruz 6C5, 1:1000), as a control for soluble fraction.

**PB2 co-immunoprecipitation**. MDCK cells were plated, infected with A/WSN-FLAG, treated with peptide, and lysed exactly as described in the previous paragraph. Lysates were then mixed with streptavidin-coated beads (Dynabeads™ M-280 Streptavidin, Thermo Fisher Scientific) and rotated for 1 h at 4 °C. The beads were subsequently washed three times for 10 min with NP-40 buffer supplemented with 0.1% Triton. In all steps, 1% BSA was included to reduce aspecific binding to the beads. After washing, the beads were heated to 95 °C in SDS-PAGE loading buffer (2% SDS) for 10 min before SDS-PAGE. An anti-FLAG antibody (anti-FLAG, Cell Signaling D6W5, 1:1000) was used to detect PB2 and the fraction that was bound to the streptavidin-coated beads was always compared with amount of

PB2 present in the total lysate ("input"). Streptavidin monomers were used a loading control for the immunoprecipitated fraction.

**Immunofluorescence**. MDCK cells were plated at a density of 30,000 cells/well in an eight chamber coverslide (155409, Lab-Tek II, Nunc) 1 day before the experiment. Cells were treated with 10 μM FITC-labeled peptide for 2 h and infected with A/WSN-FLAG (MOI = 1). After 16 h, cells were washed with PBS and fixed by incubation with 4% paraformaldehyde for 30 min. Next, cells were blocked and permeabilized with blocking buffer (PBS containing 1% BSA and 0.2% Triton X-100) and incubated with primary antibody (anti-FLAG, Cell Signaling D6W5, 1:250) for 1 h. After washing with PBS, cells were incubated with goat anti-rabbit antibody conjugated to AlexaFluor-594 for 1 h (Thermo Fisher scientific, 1:2000 in blocking buffer). After two washing steps, cells were mounted with DAPI-containing mounting medium (ProLong Gold Antifade with DAPI, Thermo Fisher scientific). Confocal images were taken using a Leica SP8 DMI8 with a HC PL APO CS2 63×/1.40 OIL objective using a HyD detector. Images were taken in uni-directional scanning mode with a line averaging of 4 and a frame averaging of 2 and a maximum laser power of 2%. Samples were visualized using a 405-diode and OPSL 488 and OPSL 552 lasers. Images were deconvolved with Huygens Suite version 16.10 (Scientific Volume Imaging, The Netherlands) with best resolution settings and adjusted for the used mounting medium (Prolong Gold, Thermo Fisher Scientific) in the LASX software. Images were cropped in Adobe Photoshop and aligned using Adobe Illustrator.

**In vitro cross-seeding experiments**. Aβ (1–42) or amylin (IAPP) were prepared as described above. For every cross-seeding experiment, 10 μM of the aggregating protein was mixed with 10% molar ratio of the peptide or preaggregated protein and 25 μM Th-T was added. This was performed in a final volume of 100 μL in Costar 3880 plates (black, nontreated, half area with transparent bottom, Corning), sealed with a transparent film. The aggregation kinetics were monitored by Th-T fluorescence at 480 nm after excitation at 440 nm. Data points were recorded every 5 min for a total time of 10 h. These experiments were all done at room temperature.

**Peptide transfection (prion seeding)**. For exposure of cells to peptides, L929 15.9 and CAD5 cells were seeded 1 day before transfection at a density of 30,000 cells per well in 24-well plates. The next day, cells were exposed to 10 or 1 μM peptides mixed with or without 1 μL Lipofectamine 2000 in 200 μL OptiMEM medium without FCS and antibiotics for 5 h. Subsequently, 300 μL growth medium was added. Cells were incubated with peptides for 3 days and confluent cell monolayers were expanded and passaged until passage 8. Cells passages 6–8 were tested for PrP$^{Sc}$ and PrP$^{C}$ content. Ten percent scrapie brain homogenates (22L, positive control) and uninfected brain homogenates (Mock, negative control) were prepared in OptiMEM using a Dounce homogenizer (Homogenisator potter S, Sartorius, Goettingen, Germany). Debris was pelleted (870 × g, 5 min, 4 °C) and supernatant was stored at −80 °C. Cells in 24-well plates were exposed to 1% v/v brain homogenate (in OptiMEM/10% FCS) from Mock mice or mice terminally sick upon infection with mouse-adapted prion strain 22L for 5 h, then the brain homogenate was diluted 1:3 with growth medium. The next day the brain homogenate was discarded and cells were subsequently cultured in medium without brain homogenate. Confluent cell monolayers were expanded and passaged up to eight times. Cells passages 6 and 8 were tested for PrP$^{Sc}$ and PrP$^{C}$ content.

**Western blot analysis and proteinase K (PK) treatment**. For PrP$^{Sc}$ detection, cells were lysed in lysis buffer (100 mM NaCl, 10 mM Tris/HCl pH 7.5, 10 mM EDTA, 0.5% Triton X-100, 0.5% desoxycholate acid sodium salt) and lysates were digested with 20 μg/mL (PK) at 37 °C for 30 min. Proteolysis was stopped by adding 0.5 mM Pefabloc (Roche). The samples were then precipitated with methanol at −20 °C and PrPSc was detected by immunoblot using anti-PrP antibody 4H11. For detection of total PrP (PrP$^{C}$ and PrP$^{Sc}$) and GAPDH, lysate aliquots were supplemented with 0.5 mM Pefabloc and precipitated with methanol. Aliquots were analyzed on NuPAGE®Novex® 4–12% Bis-Tris Protein Gels (Life Technologies) followed by transfer onto a PVDF membrane (GE Healthcare) in a wet blotting chamber overnight at 4 °C. PrP$^{C}$ blots were stripped with Re-blot solution (Merck Millipore, Darmstadt, Germany) and re-probed with an antibody directed against GAPDH ab8245 (1:1000; Abcam).

**In vivo efficacy study**. Female 7–8-week-old BALB/c mice were purchased from Charles River (Italy) and housed in individually ventilated cages under specific-pathogen-free conditions in a temperature-controlled biosafety level 2 room with 14/10-h light/dark cycles. All mouse experiments were conducted according to the National (Belgian Law 14/08/1986 and 22/12/2003, Belgian Royal Decree 06/04/2010) and European (EU Directives 2010/63/EU, 86/609/EEG) animal regulations. All protocols were approved by the Institutional Ethics Committee of Ghent University (Eth. Com. no. 2018-010). To assess the in vivo antiviral activity of selected peptides and controls, mice were treated i.v. with 100 μL of peptide solution (2.5 mg/kg) 4 h after infection and every 24 h thereafter. Infection was done intranasally under ketamine/xylazine sedation by instilling 50 μL of mouse-adapted A/PR8 or B/Mem (2x LD$_{50}$) in PBS in both nostrils. Six days after infection/treatments, mice were euthanized and lungs were aseptically removed for titration of pulmonary virus titers. For this, lungs were homogenized in 1 mL PBS using metal beads and the Qiagen TissueLyser II. Cell debris was removed by centrifugation and virus titers were determined by titrating serial dilutions in plaque assay.

**Supplementary methods**. Supportive methods such as protein synthesis, peptide synthesis, antiviral assays, and peptide radio-labeling experiments are described in Supplementary Methods.

**Statistics and reproducibility**. Statistical analysis was performed using Prism (Graphpad Version 7.0) or R-studio (Version 1.1.456). Sample sizes ($n$) of animals or number of biological repeats of experiments are indicated in the specific figures. Unpaired Student's $t$ test, one-sample $t$ test, and ANOVA were used to determine significant differences between samples unless otherwise indicated. Significance levels: $*P < 0.05$; $**P < 0.01$; $***P < 0.001$; and $****P < 0.0001$. Nonsignificant differences ($P > 0.05$) are not separately labeled, unless stated otherwise. Micrographs, confocal images, and western blots were all repeated as four independent experiments, except for Fig. 5d, which was repeated in five independent experiments. One representative image is shown in the manuscript.

**Reporting summary**. Further information on research design is available in the Nature Research Reporting Summary linked to this article.

## Data availability

All available PB2 sequences of influenza A were retrieved from fludb.org (https://www.fludb.org/brc/influenza_sequence_search_protein_display.spg?method=ShowCleanSearch&decorator=influenza). The datasets generated during and/or analyzed during the current study are available from the corresponding author on reasonable request.

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

## Acknowledgements

E.M. was supported by a PhD Fellowship from the Funds for Scientific Research Flanders (FWO). The Switch Laboratory was supported by grants from the European Research Council under the European Union's Horizon 2020 Framework Programme ERC Grant agreement 647458 (MANGO) to J.S., the Flanders institute for biotechnology (VIB), the University of Leuven ("Industrieel Onderzoeksfonds"), the Funds for Scientific Research Flanders (FWO, project grant G045920N), the Flanders Agency for innovation by Science and Technology (IWT, SBO grant 60839), the FWO project AKUL/15/34 (Hercules Grant), and the Federal Office for Scientific Affairs of Belgium (Belspo), IUAP, grant number P7/16. The Saelens Lab was supported by IWT, SBO grant 60839. Super resolution fluorescence and electron microscopy were performed at the Light Microscopy and Imaging Network (Limone) and the Electron Microscopy platform of VIB-KU Leuven, respectively. We would like to thank Mathias De Decker for taking the confocal microscopy images and Caroline Collard for co-performing the Zika screen. Part of this research work was performed using the "Caps-It" research infrastructure (project ZW13-02) that was financially supported by the Hercules Foundation (FWO) and Rega Foundation, KU Leuven.

## Author contributions

E.M. performed all biophysical experiments, the in vitro aggregation assays, the in vitro cap-binding assays, and all cellular assays. E.M. and K.R. performed all antiviral assays and all cellular infection assays. K.R. performed in vivo studies with influenza-infected mice. A.S. and L.I. performed the initial antiviral screen. R.G., J.S., and F.R. designed all peptides. M.S. performed the in vivo biodistribution experiments under supervision of G. B. Author B.H. performed the in silico analysis of all influenza A strains. R.G. and R.K. supervised biophysical experiments. R.K. co-performed all biophysical experiments. P.G. and M.R. co-performed MST experiments. Lad.K. and Lal.K. performed the entire in vivo toxicity study. H.W. purified all proteins in this study. N.L. performed the in silico APR interaction study. S.K. and J.N. performed the in vitro ZIKV screening assay. P.B. performed the cryo-EM experiments. S.L. and I.V. performed all PrP seeding experiments. E. M., J.S., and F.R. wrote the manuscript with input from all coauthors. J.S., F.R., R.G., and X.S. initiated and supervised this project.

## Competing interests

The authors declare the following competing interests: J.S. and F.R. are listed as inventors of patents held by their hold institution VIB, covering the peptides described in this manuscript, and which are licensed to Aelin Therapeutics (Leuven, Belgium), of which J.S. and F.R. are the scientific founders. All other authors declare no competing interests.
