## [Peer Review File · Nature Communications]

Reviewers' comments:

Reviewer #1 (Remarks to the Author):

In the present work, Michiels and co-workers investigate whether amyloid-virus interactions are driven by APR-APR interactions. The results of the study are well supported by the experimental data, and this work is a significant contribution to the field. They have performed a multi-faceted analysis incorporating both molecular and biophysical methods.

Minor revisions:

1. The authors should reread their manuscript in order to correct some grammatical and syntactical errors. A few sentences are quite long, making it difficult to understand their meaning.
2. The authors are advised to write the numbers of the amino acids corresponding to the ends of the peptides, wherever this is possible. For example, VGGVVI could be written as 36VGGVVI41.
3. Additionally, the numbers of the amino acids should not be in the sequence. For example, L381IQLIVS387 should be modified to 381LIQLIVS387.
4. In lines 991 and 1051, "ONE-way ANOVA" should be replaced with "one-way ANOVA".
5. Figure 2 is blurred. If this is not due to the pdf conversion process, consider increasing the size of the image.
6. All decimal numbers should be separated by a dot, not a comma.
7. There are references that have different font from the main text.
8. The authors mention that they have conducted local alignment utilizing BLAST. Have they tried an alignment with a more qualified program, like CLUSTAL Omega? And are the results the same?

Reviewer #2 (Remarks to the Author):

Previous studies have identified the amyloid protein, associated with Alzheimer's Disease (AD), can inhibit virus and bacteria. The authors used bioinformatics to design very short aggregation peptide region from the viral proteins (of influenza and Zika virus) which are homologous to human amyloid, and confirm that these short peptides have anti-flu and anti-zika-viral activity. The idea is not new. But they are first group to make such short peptides for antiviral action and they further improve them by flanking them with hydrophilic amino acids so as to improve solubility and availability. They demonstrated that 12B binds to PB2 with broad-spectrum anti-influenza A activity, and showed that the antiviral activity of this peptide can tolerate a single conservative substitution.

Major concerns:

1. The antiviral activity of the peptides shown in this study was very modest. Only plaque size reduction in cell culture assay can be achieved. And only a 2-3 fold viral load reduction in mice, not at the magnitude of log reduction. There is no survival data to prove that the antiviral peptide can salvage challenged mice.
2. The findings of the Cryo-EM used in this study is just like the TEM, and there was no structural binding analysis provided.
3. Lots of data are given but the general quality of the data are low, especially for those given in supplementary information.
4. The text is very difficult to follow and need extensive rewriting.

Individual comments:

1, Figure 1. mainly confirmed the findings of previous studies but using short peptides.

2, Supplementary Figure 2 AB and Fig S7-8, there are no error bars. All these experimental studies must be done by at least 3 independent experiments to correct for random errors.

3, There are no peptide sequences for Supplementary Figure 2 AB. We ask this question is because authors claimed in line 128,129 in results (page 6) that they supercharged the APR (including peptide 12 and therefore variants A to H) with basic arginine residues, but when we look at supplementary table 2, we found that the peptide 12 variants including 12 A,B, C and D are endowed with negative charges. This finding is completely contradictory to their claim.

4, The author claims that 12B targeted PB2. Then the authors should not select NP inhibitor and NA inhibitor as the positive control. They should select PB2 inhibitor, VX-787, as the positive control in Fig. 2C.

It is difficult to understand why did authors use the plaque size to calculate the IC50, but not used the plaque reduction assay to test IC50. The gold standard for antiviral activity is the Plaque reduction assay, the reduction of virus titre in culture supernatant of multicycle infection by cell culture and the protection against cytopathic effects. These findings MUST be provided in any antiviral studies.

5, The authors should choose an antiviral such as VX-787 that can inhibit viral replication in Fig 2G. The authors used oseltamivir which did not show any antiviral effect in the mice model. Note that oseltamivir by oral gavage should give obvious reduction in lung virus titer of treated mice after challenge. This is again contradictory and throws lots of doubt on the creditability of this study. Moreover, there was only 2-3 fold viral load reduction in lung tissue after treatment of infected mice by the 12B peptide. No survival data was given to prove the efficacy of the 12B peptide given by intravenous injection.

6, In Fig S5F, the TEM pictures indicated that the PB2 is not visualized as single particle which suggested that there are purification problem. How was PB2 protein being purified?

8, In Fig 4I, at virus inoculum of 1MOI, immunofluorescence microscopy done at 16h post-infection, why was there no PB2 visualized in 12Bpro2 treated cell and the buffer treated cells?

9, In Fig 5A and Line 286, authors claimed that sequence-specific process is tolerant to single

conservative substitutions which suggests the potential of a novel broad-spectrum antiviral. However, even though for the 100% matched LIQLIVS, the antiviral activities against the four viruses are very different with inhibition potency varies from 20% to 80% at the 10uM peptide concentration. Thus it is unlikely that when you have single substitutions, such poor antiviral potency can be maintained for all 4 different viruses.

10, Line 299, Authors claimed that 'R50 outperforming 7-DMA for anti-Zikavirus', but the IC50 and TC50 of 7-DMA are better than those of R50. This is again contradictory.

Recommendation: There are many serious shortcomings in the work that must be addressed before the paper can be considered for publication.

Reviewer #3 (Remarks to the Author):

In their paper Michiels et al. searched for sequence homologies in aggregation-prone regions in amyloidogenic proteins and viral proteins speculating that these APR regions may interact physically thereby affecting viral replication. They identified such shared APR and engineered a designer peptide derived from the Influenza A PB2 peptide. This peptide inhibited Influenza A replication in cell culture and in vivo in mice. Using the same approach, they also created a similar peptide that was shown to inhibit ZIKV infection.

Although the hypothesis is thrilling I have severe doubts regarding several parts of the manuscript:

1. Authors searched for APR in amyloids and viral proteins and found homologies assuming that possible interactions of respective regions occur also in vivo. At least in part, the some assumptions are wrong, because e.g. full length PAP (Fig 1E and S Table 1) does not interact with HIV-1 particles (only fibrils derived from PAP85-120 and PAP248-286) and the fibrils do not bind the viral glycoprotein but the membrane of virions. So, there is no viral counterpart. Whether alpha synuclein indeed interacts with specific viral proteins has to my knowledge also yet proven. As previously published by the authors and other, APR can be found in almost all proteins and buried inside. So it is not surprising that one finds similar sequences in the human and viral proteome if all database entries are screened. Are there ARP in non-amyloidogenic proteins that share similarities with virus ARPs?

2. Secondly, the 12B peptide has Influenza inhibitory activity in vitro and in vivo, but the mechanism of action is hardly explored (see more details below). As shown, 12B rapidly forms amyloid. Has the amyloid form of 12B been tested in vivo or the monomer? Moreover, most of the results may simply be explained by competition of the amyloid with endocytotic pathway route of Influenza virus (we have unpublished data that many amyloids unspecific ally interfere with Influenza virus entry by competition with amyloids and the similar mechanism may also play role in the antiviral activity of 12B. It is unclear how the peptide or the amyloid derived thereof is internalized and by which mechanism it may prevent Influenza replication. A 12B mutant not carrying an ARP might be a good control. Moreover, classical time of addition experiment need to be performed to dissect whether 12B inhibits entry or whether it acts indeed intracellularly. It is also possible the amyloid sequesters the virus rendering it non-infectious which could be tested by pre-exposing the virions to 12B first followed by infection.

3. ZIKV inhibitory peptide: Generally, this part is much less elaborated than the Flu part. Several controls on the antiviral mechanism are missing and also non-specific effects such as toxicity, supra structure formation by the peptide, binding of virions etc could account for the observed anti-ZIKV activity. Toxicity assays should be performed by incubating cells with peptide at concentrations that exceed the IC50 by 100 fold allowing to determine selectivity indexes. Time of addition experiments need to be done to test at which stage of the reoication cycle the peptide acts (and whether it is active as monomer or amyloid).

More comments:

L 36 PB2 accumulates in influenza-infected tissue in vivo and displays antiviral activity against influenza A and its common PB2 polymorphisms: rephrase ,

L50 interferors is not the correct wording, e.g. in some instances amyloids clearly enhance infection

L51 there are also studies on semenogelins derived fibrils that enhance HIV-1 infection

L42-61: Introduction would benefit from a differentiation in those amyloids that have antiviral and those having proviral activity

L79 influenza A we show that the amyloid accumulates: which amyloid is meant?

L90: Semenogelin fragments are missing;

L95 and table 1: Wrong: PAP does not interact with the viral glycoprotein, PAP derived peptides 248-286 and 85-120 form fibrils that bind the membrane of the virus but not the glycoproteins

L101: scrambled peptides (same pI and charge) would have been a better control

L111-119: I don't get the meaning of Fig. 1E. E.g. what is the red mark "HIV-1" in the PAP item meaning? Is there a homology in the first PAP residues with those in HIV-1? And if yes, against which viral proteins? It would be good to have a control like albumin and hemoglobin and do the same analysis for the indicated viruses to show that this kind of assay yields meaningful results.

L124 why not using an APR from the better established A β - HSV link?

L126 why wasn't the APR as predicted synthesized but rather a designer peptide (dimer with linker) What is the rationale?

L138: show hemolytic activity or cytotoxicity assay of peptide 12, the antiviral effect could be caused by toxicity.

L 144: negatively charged, is it just a polyanion if forming fibrils?

Fig. 3F. How has the cytotoxic assay been performed? How many days were the cells exposed to the peptide before toxicity was determined?

Comment on solubility of the peptide?

Typically in vivo experiments are at the end? In which state was 12B applied? As monomer or amyloid?

Line 182: the peptide is already present in its amyloid state. Has the amyloid been administered in vivo?

Line 188: 3D does not show classical amyloid fibrils it looks like oligomers? What does the inlet 100 μ M mean?

Fig. 3G: It would be much more convincing to titrate the peptides to see whether there is indeed activity or not as a general property of the peptides, or just an effect obtained at a certain concentration. The antiviral assays are easy to perform.

Line 219: Why is the scrambled peptide binding?

Line 271: same MOI of Influenza A and B viruses used? Mutational analyses of PB2 in Flu A would be more convincing

Line 293: provide sequences as table.

Lines 311-325: many of the mentioned amyloid/virus interaction are only indirect (no physical interaction between amyloid and the viral structures); and sometimes data were only derived from in vitro experiments when amyloid (eg IAPP) was agitated with virus (e.g. RSV). As outlined controls should be included like albumin or hemoglobin subunits as searched for APRs and sequence similarities.

The discussion would benefit from a much more detailed explanation of the observed anti-Flu effects. How is the peptide internalized? Is it internalized as amyloid or soluble peptide? How does the peptide get into contact with PB2 in the cell? What is the mechanism by which it blocks Flu replication? How many copies of the peptide per cell are required to block replication? Stoichiometric or substoichiometric amounts required? Is there amyloid stainable inside the cell?

Reviewer #1 (Remarks to the Author):

In the present work, Michiels and co-workers investigate whether amyloid-virus interactions are driven by APR-APR interactions. The results of the study are well supported by the experimental data, and this work is a significant contribution to the field. They have performed a multi-faceted analysis incorporating both molecular and biophysical methods.

Minor revisions:

1. The authors should reread their manuscript in order to correct some grammatical and syntactical errors. A few sentences are quite long, making it difficult to understand their meaning.

This is adjusted in the revised manuscript. Overall, we simplified the language that is used and reformulated some complex concepts.

2. The authors are advised to write the numbers of the amino acids corresponding to the ends of the peptides, wherever this is possible. For example, VGGVVI could be written as 36VGGVVI41.

This is adjusted in the revised manuscript. Briefly, when the peptides refer to a part of a protein we included this numbering as suggested by Reviewer #1. However, when we refer to a peptide as an independent molecule, we did not add this numbering to avoid confusion.

3. Additionally, the numbers of the amino acids should not be in the sequence. For example, L381IQLIVS387 should be modified to 381LIQLIVS387.

This is adjusted in the revised manuscript.

4. In lines 991 and 1051, "ONE-way ANOVA" should be replaced with "one-way ANOVA".

This is adjusted in the revised manuscript.

5. Figure 2 is blurred. If this is not due to the pdf conversion process, consider increasing the size of the image.

At this stage we indeed included low-quality figures to compress the file into one document. High quality images are included in the revised manuscript.

6. All decimal numbers should be separated by a dot, not a comma.

This is adjusted in the revised manuscript.

7. There are references that have different font from the main text.

This is adjusted in the revised manuscript.

8. The authors mention that they have conducted local alignment utilizing BLAST. Have they tried an alignment with a more qualified program, like CLUSTAL Omega? And are the results the same?

We used the local alignment tool BLAST only to align two sequences, not to perform multiple sequence alignments. CLUSTAL Omega is mainly used to perform alignments of three or more sequences. For our multiple alignments of all the influenza A strains, we used

MAFFT, a qualified multiple alignment tool that overall is the superior tool, even compared with CLUSTAL Omega (Wilm, Mainz, & Steger, 2006).

Reviewer #2 (Remarks to the Author):

Previous studies have identified the amyloid protein, associated with Alzheimer's Disease (AD), can inhibit virus and bacteria. The authors used bioinformatics to design very short aggregation peptide region from the viral proteins (of influenza and Zika virus) which are homologous to human amyloid, and confirm that these short peptides have anti-flu and anti-zika-viral activity. The idea is not new. But they are first group to make such short peptides for antiviral action and they further improve them by flanking them with hydrophilic amino acids so as to improve solubility and availability. They demonstrated that 12B binds to PB2 with broad-spectrum anti-influenza A activity, and showed that the antiviral activity of this peptide can tolerate a single conservative substitution.

Our synthetic amyloids are designed solely based on the primary amino acid information encoded into the viral proteins. In other words, our synthetic amyloids are not at all "homologous to human amyloid" and do not occur in nature. Unfortunately, Reviewer #2 feels that this is not new, which clearly shows that (s)he did not fully appreciate the novelty of the design principles for antiviral activity described in the current manuscript. The sole observation that has been made so far is that an amyloid-forming peptide associated with Alzheimer's disease, called 'amyloid beta', interferes with herpes virus replication by a physical interaction with the virion particles. We use this interesting observation as a starting point in our manuscript to explore whether we can exploit the intrinsic aggregation propensity of viral proteins to design **synthetic amyloids** that can specifically bind to the viral protein they are based on (in this case influenza and zika proteins).

Below, Reviewer #2 also mentions that the text is difficult to follow. We agree and have substantially altered the manuscript to more clearly emphasize novelty. We now clearly indicate we are not using any known human amyloids in our synthetic design and therefore removed the distraction of the data from Figure 1 which describes natural amyloid-virus interactions. We thus now solely focus on how we design, de novo and from sequence information only, synthetic amyloids that show antiviral properties.

Major concerns:

1. The antiviral activity of the peptides shown in this study was very modest. Only plaque size reduction in cell culture assay can be achieved. And only a 2-3 fold viral load reduction in mice, not at the magnitude of log reduction. There is no survival data to prove that the antiviral peptide can salvage challenged mice.

In addition to the plaque size reduction assay we included two additional experiments in the revised manuscript to confirm *in vitro* antiviral activity: multicycle infection and cytopathic effects (**Supplementary Fig. 1g-1h** and **Supplementary Fig. 6b-6c**). These additional results confirm the specific antiviral activity of our peptides *in vitro*. Although this antiviral effect observed *in vitro* is strong and very consistent, the *in vivo* activity is indeed modest. However, as highlighted in the manuscript, we were able to pinpoint this problem. By the design of a radio-labeled variant, we showed that our synthetic amyloid has a half-life of ~15 minutes. Since we perform daily injections in these mice (more injections are difficult to handle for these animals), it would be very unlikely to obtain large *in vivo* effects on a virus replicating at an exponential rate. That being said, we did include all the proper controls

(including control peptides) to show that the relatively modest effect we observe is nonetheless real and significant (**Fig. 1g**). The goal of the current paper however is not drug development, but to demonstrate proof of concept for a novel antiviral strategy and its potential *in vivo*. Developing a superior new antiviral drug is beyond the scope of the current work.

2. The findings of the Cryo-EM used in this study is just like the TEM, and there was no structural binding analysis provided.

We respectfully disagree with the Reviewer's interpretation of the cryo-EM data that we provided. As mentioned in the manuscript, the sole goal of using additional cryo-EM experiments was to validate that the amyloid structures, as seen in regular TEM experiments, also form in solution and are not an artefact of the drying step performed in these regular TEM experiments, or the staining protocols employed. There was never an intention, nor a need, to provide any structural binding analysis on these amyloids and the fact that the cryo-EM images resemble those of the regular TEM is in fact the positive outcome of this assay.

3. Lots of data are given but the general quality of the data are low, especially for those given in supplementary information.

We are very disappointed by this negative comment. We are happy to revise or even remove figures that are of low quality, but Reviewer #2 does not give a valid reason, nor specifies which data is of low quality.

4. The text is very difficult to follow and need extensive rewriting.

We apologize for this problem, which clearly set this Reviewer on the wrong foot. To solve this issue, we rewrote and clarified a large portion of the manuscript. As highlighted above, we also removed the data presented in Figure 1 to avoid sidetracking and to provide focus on our novel approach towards developing antiviral peptides.

Individual comments:

1. Figure 1. mainly confirmed the findings of previous studies but using short peptides.

It was previously shown that human amyloids, including amyloid beta, can indeed interact with viral particles. This figure was used as initial evidence that short APRs could be sufficient to drive an interaction with a virus and is the inspiration that formed the starting point for the design of our **fully synthetic amyloids**, solely based on an APR identified in a viral protein. However, as highlighted above, we removed the data presented in Figure 1 to allow a better understanding of our new concept. We also shifted the focus towards the design of synthetic antiviral amyloids and highlighted the mode-of-action.

2. Supplementary Figure 2 AB and Fig S7-8, there are no error bars. All these experimental studies must be done by at least 3 independent experiments to correct for random errors.

These are our initial screening assays. From this data, we do not conclude whether a specific peptide is active or not. We use this data to select the most promising peptide and only make **statistically correct statements** in the repeat experiments described afterwards. We agree with the reviewer that the set might contain false negatives or positives, but it was

not our purpose to identify an exhaustive list of effective sequences, merely to identify a small number of hits that could be investigated further. The subsequent experiments also show that for the selected peptides the effect observed was not due to random errors and indeed reduce viral replication significantly.

3. There are no peptide sequences for Supplementary Figure 2 AB. We ask this question is because authors claimed in line 128,129 in results (page 6) that they supercharged the APR (including peptide 12 and therefore variants A to H) with basic arginine residues, but when we look at supplementary table 2, we found that the peptide 12 variants including 12 A,B, C and D are endowed with negative charges. This finding is completely contradictory to their claim.

For clarity, all peptide sequences are now included in the manuscript (**Supplementary Table 1 and 4**), however, the reviewer appears to be confused as to our claim. **Our design strategy is not limited to positively charged arginine residues, nor do we state this anywhere in the manuscript.** The word supercharged is correct, since the net charge of all peptides is 4. However, the sign of the charge does vary between peptides: In line 128 (now: 88) it is indeed mentioned that the initial screen is done with arginine residues to flank the APRs (**Suppl Fig 1a-1b**). As a result, peptide 12 is indeed capped with positively charged arginine residues. However, in lines 135-146 (now: 99-109) it is also clearly stated that, following this initial screen, the APR of peptide 12 was selected and its **design was modified** to optimize the antiviral effect by **screening “multiple charged amino acids”** (and other variables). As a result, peptide variants 12A-12D are capped with negatively charged aspartate residues and 12E-12H are capped with positively charged arginine residues (**Supplementary Table 2**). Again, this does not contradict in any way with our claims. Also, the comment of Reviewer #2 referred to line 128, while the answer was located in line 135-146.

4. The author claims that 12B targeted PB2. Then the authors should not select NP inhibitor and NA inhibitor as the positive control. They should select PB2 inhibitor, VX-787, as the positive control in Fig. 2C.

We used two specific positive controls in **Fig 1c-1d**, Nucleozin and Tamiflu, an NP and NA inhibitor, respectively, with specific intentions. Tamiflu is the most commonly used anti-influenza compound so using this compound allows comparison to many anti-influenza studies. Nucleozin was selected specifically for its mechanism of action: it induces NP aggregation, so it is the closest possible match to our mode-of-action (Kao et al., 2010). VX-787 is a rarely used compound, however, we included this control in the revised manuscript as suggested by Reviewer #2 (**Fig 1c-1d**).

5. It is difficult to understand why did authors use the plaque size to calculate the IC50, but not used the plaque reduction assay to test IC50. The gold standard for antiviral activity is the Plaque reduction assay, the reduction of virus titre in culture supernatant of multicycle infection by cell culture and the protection against cytopathic effects. These findings **MUST** be provided in any antiviral studies.

We used the **plaque-size reduction assay** to determine the IC50 of our peptides. Moreover, we deliberately used the plaque-size reduction assay throughout the paper in all influenza assays to reach consistency. However, as suggested by Reviewer #2, we included two additional *in vitro* assays in the revised manuscript: the multicycle infection assay and the

protection against cytopathic effects (**Supplementary Fig. 1g-1h** and **Supplementary Fig. 6b-6c**). These assays convincingly show the antiviral effect of our synthetic amyloid.

6. The authors should choose an antiviral such as VX-787 that can inhibit viral replication in Fig 2G. The authors used oseltamivir which did not show any antiviral effect in the mice model. Note that oseltamivir by oral gavage should give obvious reduction in lung virus titer of treated mice after challenge. This is again contradictory and throws lots of doubt on the creditability of this study. Moreover, there was only 2-3 fold viral load reduction in lung tissue after treatment of infected mice by the 12B peptide. No survival data was given to prove the efficacy of the 12B peptide given by intravenous injection.

This is not correct. It is well-described that oseltamivir has no or very limited effect on lung virus titers of mouse-adapted PR8 strains (Byrn et al., 2015; Chockalingam et al., 2016). Even more, we use roughly the same virus dose to infect mice as described in those manuscripts and observe the same effect. In short, our findings are in complete agreement with current literature. As mentioned above, we did include all the proper controls (including control peptides) to show that the **modest effect we observe *in vivo* is indeed valid**. Instead of developing a superior new antiviral drug, we here show a new concept of synthetic antiviral amyloids and did not anticipate a massive *in vivo* efficacy.

7. In Fig S5F, the TEM pictures indicated that the PB2 is not visualized as single particle which suggested that there are purification problem. How was PB2 protein being purified?

We have purified PB2 using state of the art techniques, in which we have ample expertise and experience. We have extensive Dynamic Light Scattering data to show that PB2 is not aggregated at the start of our experiments. This data is included in the revised manuscript (**Supplementary Fig. 4a**). However, it is expected that this protein aggregates to some extent when it is incubated at 30°C for 24 hours at a concentration of 32 μ M (as described in the methods section). This aggregation results in the structures observed in the TEM experiment. Finally, the full purification process is described in detail in the methods section.

8. In Fig 4I, at virus inoculum of 1MOI, immunofluorescence microscopy done at 16h post-infection, why was there no PB2 visualized in 12Bpro2 treated cell and the buffer treated cells?

At this stage of the manuscript submission process, we included low-quality figures to compress the file into one document. The PB2 is definitely visible in the 12Bpro2-treated cells, however, not in large inclusions as seen in 12B-treated cells. High quality images are used in the revised manuscript, so PB2 will be properly visible. To clarify, this data shows that when 12B is present, PB2 aggregates into inclusions, while this is not the case for 12Bpro2, in which PB2 shows the same diffuse distribution as in buffer-treated cells.

9. In Fig 5A and Line 286, authors claimed that sequence-specific process is tolerant to single conservative substitutions which suggests the potential of a novel broad-spectrum antiviral. However, even though for the 100% matched LIQLIVS, the antiviral activities against the four viruses are very different with inhibition potency varies from 20% to 80% at the 10 μ M peptide concentration. Thus it is unlikely that

when you have single substitutions, such poor antiviral potency can be maintained for all 4 different viruses.

With this experiment, we demonstrate the principle that our approach restricts replication of seven different influenza A viruses, covering different subtypes and host species. The observed restriction efficiencies indeed varied dependent on the virus but in all cases were statistically significantly different from the control treatment. Moreover, there was no effect on any of the 3 influenza B viruses tested. The variable efficiency of the inhibition is likely due to different entry, uncoating and replication kinetics of the tested viruses *in vitro*, parameters that we did not compare or optimized the 12B peptide concentration for.

10. Line 299, Authors claimed that 'R50 outperforming 7-DMA for anti-Zikavirus', but the IC₅₀ and TC₅₀ of 7-DMA are better than those of R50. This is again contradictory. In our initial screening assay, amyloid R50 performed better compared to 7-DMA, as can be seen in **Supplementary Fig. 8-9** (this is where we used the wording 'outperformed'). However, in the detailed IC₅₀ assay, both IC₅₀s do not differ significantly from one another. In the revised manuscript we rephrased our initial statement, as suggested by Reviewer #2.

Recommendation: There are many serious shortcomings in the work that must be addressed before the paper can be considered for publication.

Reviewer #3 (Remarks to the Author):

In their paper Michiels et al. searched for sequence homologies in aggregation-prone regions in amyloidogenic proteins and viral proteins speculating that these APR regions may interact physically thereby affecting viral replication. They identified such shared APR and engineered a designer peptide derived from the Influenza A PB2 peptide. This peptide inhibited Influenza A replication in cell culture and in vivo in mice. Using the same approach, they also created a similar peptide that was shown to inhibit ZIKV infection.

Although the hypothesis is thrilling I have severe doubts regarding several parts of the manuscript:

1. Authors searched for APR in amyloids and viral proteins and found homologies assuming that possible interactions of respective regions occur also in vivo. At least in part, some assumptions are wrong, because e.g. full-length PAP (Fig 1E and S Table 1) does not interact with HIV-1 particles (only fibrils derived from PAP85-120 and PAP248-286) and the fibrils do not bind the viral glycoprotein but the membrane of virions. So, there is no viral counterpart. Whether alpha synuclein indeed interacts with specific viral proteins has to my knowledge also yet proven. As previously published by the authors and other, APRs can be found in almost all proteins and buried inside. So, it is not surprising that one finds similar sequences in the human and viral proteome if all database entries are screened. Are there APR in non-amyloidogenic proteins that share similarities with virus APRs?

It is correct that in most of our examples no physical interaction has been shown between the human-infecting virus and the human amyloid, herpes and amyloid beta being the only exception, rather than clinical correlations between increased amyloid levels and viral infections. The sole goal of Figure 1 was to introduce the concept of our approach and highlight that in natural context those homologous APRs are at the least present between the virus and the amyloid. We did not intent to draw any conclusions from this, but merely introduce our novel concept: APR-based synthetic amyloids with antiviral properties. Therefore, to reduce the complexity of our manuscript, we removed the data of Figure 1 so future readers can focus on the novel findings regarding APR-based antiviral amyloids. Instead, we clarified the aspects of the only known physical interaction between an amyloid and a virus (amyloid beta and herpes virus) in the revised introduction.

2. Secondly, the 12B peptide has Influenza inhibitory activity in vitro and in vivo, but the mechanism of action is hardly explored (see more details below). As shown, 12B rapidly forms amyloid. Has the amyloid form of 12B been tested in vivo or the monomer? Moreover, most of the results may simply be explained by competition of the amyloid with endocytotic pathway route of Influenza virus (we have unpublished data that many amyloids unspecific ally interfere with Influenza virus entry by competition with amyloids and the similar mechanism may also play role in the antiviral activity of 12B. It is unclear how the peptide or the amyloid derived thereof is internalized and by which mechanism it may prevent Influenza replication. A 12B mutant not carrying an APR might be a good control. Moreover, classical time of addition experiment need to be performed to dissect whether 12B inhibits entry or whether it acts indeed intracellularly. It is also possible the amyloid sequesters the

virus rendering it non-infectious which could be tested by pre-exposing the virions to 12B first followed by infection.

We thank Reviewer #3 for pointing out that we need to clarify this part, since the mechanism of action is the most innovative part of this manuscript. We already obtained substantial amount of data showing a large part of the mechanism of action. In short, we showed that our peptide rapidly organizes into amyloids after solvation (**Fig. 2a-2h**). As a result, all experiments performed in the manuscript, also *in vivo*, include the use of the amyloid state of 12B. Furthermore, we showed that amyloid 12B enters cells (**Fig. 4c**), specifically interacts with PB2 in the cell (**Fig. 4c-4d**) and induces PB2 aggregation as seen from the loss of soluble PB2, eventually leading to a reduced viral replication (**Fig. 4e**). Reviewers #3 also raises her/his concern regarding aspecific effects of an amyloid on endocytic pathways. We anticipated this by designing a control peptide that also organizes into amyloids (12Bscr2, **Supplementary Fig. 3**). This amyloid did not show any effect on viral replication in all of the assays tested (**Fig. 2i, Fig. 4d-4e**), showing that in our assays, amyloids alone do not interfere with influenza replication. Reviewer #3 suggests to include a 12B mutant not carrying an APR, which is exactly the control we already included, namely 12Bpro2 (**Supplementary Fig. 3**). This peptide also does not affect viral replication in any assay. Finally, Reviewer #3 gives the very interesting suggestion to perform the typical time of addition experiments and an experiment where virions are pre-exposed to the amyloid before infection to provide even more insights into the antiviral mechanism of our amyloids. We included these experiments in the revised manuscript and believe that they are indeed a major contribution to the elucidation of the mechanism of action of our synthetic amyloids. The time of addition experiment shows that the antiviral effect of our amyloid does not change, even if it is added 6 hours after virus infection (**Fig. 4a, Supplementary Fig. 5a**). Additionally, no effect is seen on influenza B, showing that aspecific interference with viral entry is not our mechanism (**Supplementary Fig. 5a**). The pre-treatment of virions was also performed and shows that the effect is roughly the same compared to our regular method (**Fig. 4b, Supplementary Fig. 5b**). However, again no effect is observed when influenza B virions are pre-incubated with our amyloid (**Supplementary Fig. 5b**), **excluding the possibility of an aspecific effect that would block virus entry**. Instead, our amyloid acts intracellularly by a specific binding event to PB2, as can be seen in the immunostainings in **Fig. 4c**. A description of these results and the mechanism of action of our amyloids is provided in the revised discussion.

3. ZIKV inhibitory peptide: General, this part is much less elaborated than the Flu part. Several controls on the antiviral mechanism are missing and also non-specific effects such as toxicity, supra structure formation by the peptide, binding of virions etc could account for the observed anti-ZIKV activity. Toxicity assays should be performed by incubating cells with peptide at concentrations that exceed the IC50 by 100 fold allowing to determine selectivity indexes. Time of addition experiments need to be done to test at which stage of the reoication cycle the peptide acts (and whether it is active as monomer or amyloid).

We want to emphasize that the ZIKV part was solely to proof the generality of our approach. It does not add to the mechanistic insights provided for the influenza part. Nevertheless, we included all additional experiments, as requested by Reviewer #3. First of all, we performed additional toxicity assays for R50, using a maximum concentration of 500 μ M. Only at the highest concentration, minimal toxic effects are observed (**Fig. 6i**). The revised manuscript

now also includes a full biophysical study, showing that peptide R50 readily organizes into amyloid structures (Fig. 6a-6d). In other words, as for peptide 12B, peptide R50 is used in the amyloid state in all described assays. Finally, a time-of-addition experiment is added in the revised manuscript, showing that peptide R50 remains active, even when it is added 6 hours after infection (Supplementary Fig. 11a-11b).

More comments:

1. L 36 PB2 accumulates in influenza-infected tissue in vivo and displays antiviral activity against influenza A and its common PB2 polymorphisms: rephrase.

This is adjusted in the revised manuscript.

2. L50 interferors is not the correct wording, e.g. in some instances amyloids clearly enhance infection

We removed this part in the revised manuscript.

3. L51 there are also studies on semenogelins derived fibrils that enhance HIV-1 infection

We removed this part in the revised manuscript.

4. L42-61: Introduction would benefit from a differentiation in those amyloids that have antiviral and those having proviral activity

We removed this part in the revised manuscript.

5. L79 influenza A we show that the amyloid accumulates: which amyloid is meant?

The designer amyloid, so the amyloids formed by peptide 12B. This is clarified in the revised manuscript.

6. L90: Semenogelin fragments are missing.

We removed this part in the revised manuscript.

7. L95 and table 1: Wrong: PAP does not interact with the viral glycoprotein, PAP derived peptides 248-286 and 85-120 form fibrils that bind the membrane of the virus but not the glycoproteins

We removed this part in the revised manuscript.

8. L101: scrambled peptides (same pI and charge) would have been a better control

We removed this part in the revised manuscript.

9. L111-119: I don't get the meaning of Fig. 1E. E.g. what is the red mark "HIV-1" in the PAP item meaning? Is there a homology in the first PAP residues with those in HIV-1? And if yes, against which viral proteins? It would be good to have a control like albumin and hemoglobin and do the same analysis for the indicated viruses to show that this kind of assay yields meaningful results.

We removed this part in the revised manuscript.

10. L124 why not using an APR from the better established A β - HSV link?

Our goal is not to prove or disprove any interactions that occur between viruses and amyloids in nature. We aimed to design our own amyloids targeting any possible virus. Influenza is a relatively easy model virus with well-established assays available, which is why we initially chose to design new antiviral amyloids against this specific virus.

11. L126 why wasn't the APR as predicted synthesized but rather a designer peptide (dimer with linker) What is the rationale?

Our intended mechanism of action is induced protein aggregation. It is well established that the rate-limiting step in the aggregation process is the initial engagement of two identical APRs, generating a so-called 'seed'. Our proprietary dimer design scaffold forces the engagement of two APRs, which increases its aggregation induction potential (unpublished data). Furthermore, the charged amino acids are added to reduce self-aggregation of our peptides. To clarify, a concise explanation of this concept is added in the revised manuscript (line 89-92).

12. L138: show hemolytic activity or cytotoxicity assay of peptide 12, the antiviral effect could be caused by toxicity.

This data is added in the revised manuscript in **Supplementary Fig. 1c**.

13. L144: negatively charged, is it just a polyanion if forming fibrils?

Our peptide 12B is indeed negatively charged, however, as well as all our control peptides, also the one that forms amyloid (12Bscr2).

14. Fig. 3F. How has the cytotoxic assay been performed? How many days were the cells exposed to the peptide before toxicity was determined?

The cells were exposed to the peptide for 24 hours. A more detailed description is added in the revised manuscript (**Fig. 1f**).

15. Comment on solubility of the peptide?

A full biophysical analysis for all the peptides is performed (**Fig. 2** and **Supplementary Fig. 3**). However, in the revised manuscript we also included a more accurate solubility assay as requested by Reviewer #3. The peptides were incubated for indicated time periods, followed by ultracentrifugation and concentration determination of the soluble fraction (**Fig. 2h** and **Supplementary Fig. 3f**). This assay shows that a large fraction of these peptide stays soluble at the working concentration of 10 μ M.

16. Typically, in vivo experiments are at the end? In which state was 12B applied? As monomer or amyloid?

As described above, in all assays, peptide 12B is used as an amyloid. Since the mechanistic insights of our synthetic amyloids are the crucial findings of this study, we chose to highlight these findings in the end of the manuscript, rather than the *in vivo* activity.

17. Line 182: the peptide is already present in its amyloid state. Has the amyloid been administered in vivo?

Yes.

18. Line 188: 3D does not show classical amyloid fibrils it looks like oligomers? What does the inlet 100 μM mean?

Correct. These structures look like short amyloids, but since oligomers are poorly defined it is impossible to refer to these structures as actual oligomers. It is clear, however, that these structures are amyloids, since they bind the amyloid-specific dye Th-T and have the typical beta-sheet content according to FTIR (**Fig. 2b-2d**). As mentioned in the main text, the 100 μM inlet refers to the concentration of peptide used. 100 μM is the **stock concentration** in all other assays, while 10 μM is the working concentration, unless stated otherwise. This is now added in the figure legend.

19. Fig. 3G: It would be much more convincing to titrate the peptides to see whether there is indeed activity or not as a general property of the peptides, or just an effect obtained at a certain concentration. The antiviral assays are easy to perform.

We included titration experiments of our two control peptides in the revised manuscript (**Supplementary Fig. 3g-3h**). This assay indicates that these peptides do not show any antiviral activity against influenza A for a concentration up to 100 μM .

20. Line 219: Why is the scrambled peptide binding?

Since we use aggregation-prone peptides, they can be 'sticky' and bind aspecifically to other proteins. This is also the reason why we included this control peptide. The main goal of showing these K_D 's is to indicate that peptide 12B binds a lot stronger to its target protein compared to the scrambled control peptide.

21. Line 271: same MOI of Influenza A and B viruses used? Mutational analyses of PB2 in Flu A would be more convincing

This is correct, we used the same MOI for both influenza A and B in this assay. Our data show that 1 mutation is not sufficient to halt the interaction of amyloid peptide 12B with influenza A (**Fig. 5a**). Additionally, we show that the aggregation-prone protein PrP is not affected by amyloid peptide 12B, while it only contains 2 mutations (**Fig. 5h-5j**). Moreover, influenza B, which has 6 mutations in this fragment, is not affected by amyloid peptide 12B (**Fig. 5a-5b**). Finally, as suggested by Reviewer #3, we included an extensive *in silico* analysis of the predicted interaction between our synthetic amyloid and the viral target protein PB2, including up to 2 mutations (**Supplementary Fig. 7, Supplementary text**). This analysis shows that inserting 2 mutations in the PB2 target APR has a strong negative effect on the interaction potential of our amyloid. Moreover, the cases where a double mutation still allows interaction with the amyloid destabilize the full PB2 protein and hence do not occur in a natural context. Altogether, these data and *in silico* analysis indicate that our proposed mechanism is sequence specific and only allows a minimal window for mutations.

22. Line 293: provide sequences as table.

This is included in the revised manuscript, together with the sequences for the influenza screen, as suggested by reviewer #2 (**Supplementary Table 1 and 4**).

23. Lines 311-325: many of the mentioned amyloid/virus interaction are only indirect (no physical interaction between amyloid and the viral structures); and sometimes data were only derived from in vitro experiments when amyloid (eg IAPP) was

agitated with virus (e.g. RSV). As outlined controls should be included like albumin or hemoglobin subunits as searched for APRs and sequence similarities.

This again refers to Figure 1, which we removed in the revised manuscript.

24. The discussion would benefit from a much more detailed explanation of the observed anti-Flu effects. How is the peptide internalized? Is it internalized as amyloid or soluble peptide? How does the peptide get into contact with PB2 in the cell? What is the mechanism by which it blocks Flu replication? How many copies of the peptide per cell are required to block replication? Stoichiometric or substoichiometric amounts required? Is there amyloid stainable inside the cell?

Together with the additional data that is provided in the revised manuscript, we added a more detailed explanation of our mechanism of action in the discussion (lines 309-320). As mentioned above, our peptides readily organize into amyloids and are used in the amyloid form in all assays. Amyloid peptide internalization was studied by our lab before (Couceiro et al., 2015) and indicates that our small acidic amyloids are taken up by nonspecific endocytosis. In the revised manuscript we also showed that our amyloids do not aspecifically block viral entry. Instead, we shown that amyloid peptide 12B physically interacts with its PB2 target protein in the cytosol (**Fig. 4c-4d**). We hypothesize that this interaction occurs during viral protein translation, since APRs might be more solvent exposed during translation compared to a fully folded protein. We show that amyloid peptide 12B induces PB2 aggregation in the cytosol (**Fig. 4e**), by which the protein loses its function. Since the essential cap-snatching activity of PB2 is lost, viral replication is halted. Finally, it would be very challenging to determine how may copies of the peptide are needed per cell to block replication. First, we don't know how many monomeric peptides organize into one stable amyloid structure. Second, the peptide does not organize into a homogenous mixture of amyloid structures (**Fig. 2e-2f**). Thirdly, if we were able to determine the above, it would still be very challenging to determine how many of these amyloid structures are present in one individual cell.

REFERENCES

- Byrn, R. A., Jones, S. M., Bennett, H. B., Bral, C., Clark, M. P., Jacobs, M. D., et al. (2015). Preclinical activity of VX-787, a first-in-class, orally bioavailable inhibitor of the influenza virus polymerase PB2 subunit. *Antimicrobial Agents and Chemotherapy*, 59(3), 1569–1582. <http://doi.org/10.1128/AAC.04623-14>
- Chockalingam, A. K., Hamed, S., Goodwin, D. G., Rosenzweig, B. A., Pang, E., Boyne, M. T., & Patel, V. (2016). The Effect of Oseltamivir on the Disease Progression of Lethal Influenza A Virus Infection: Plasma Cytokine and miRNA Responses in a Mouse Model. *Disease Markers*, 2016, 9296457–12. <http://doi.org/10.1155/2016/9296457>
- Couceiro, J. R., Gallardo, R., De Smet, F., De Baets, G., Baatsen, P., Annaert, W., et al. (2015). Sequence-dependent internalization of aggregating peptides. *The Journal of Biological Chemistry*, 290(1), 242–258. <http://doi.org/10.1074/jbc.M114.586636>
- Kao, R. Y., Yang, D., Lau, L.-S., Tsui, W. H. W., Hu, L., Dai, J., et al. (2010). Identification of influenza A nucleoprotein as an antiviral target. *Nature Publishing Group*, 28(6), 600–605. <http://doi.org/10.1038/nbt.1638>
- Wilm, A., Mainz, I., & Steger, G. (2006). An enhanced RNA alignment benchmark for sequence alignment programs. *Algorithms for Molecular Biology : AMB*, 1(1), 19–11. <http://doi.org/10.1186/1748-7188-1-19>

REVIEWERS' COMMENTS:

Reviewer #3 (Remarks to the Author):

Authors have addressed most of my concerns and performed additional experiments.

Reviewer #4:

This reviewer submitted a confidential document, which highlights some remaining points based on your response to the points raised by reviewer #2:

Point 3: While you now explain that Aspartic acids were added, please explain why adding these negative charges is beneficial. Also, when you first describe peptide 12, please explain in which functional domain of the IAV PB2 this sequence is lying. Also, specify what is the corresponding sequence in influenza B.

Point 5: The reviewer agrees with reviewer 2 that two-fold reduction in PFU (lung virus titer) is minimal, and it is very hard to believe that this effect was significant. If you want to keep the in vivo results, you should use much more subtle wording and explain that the in vivo effect was 'marginal'. [Editorially we think that it is okay that the synthetic amyloid is not quite ready for therapeutic demonstration in vivo, and ask that you caveat and walk back this claim.

All other points were judged to be addressed, or editorially overruled. There was a request to use VX787 as a positive control in Fig 3f, but as additional experiments at this time are not feasible and the absence of this control will not alter the overall message, we decided that it was not necessary.

REVIEWERS' COMMENTS:

Reviewer #4:

This reviewer submitted a confidential document, which highlights some remaining points based on your response to the points raised by reviewer #2:

Point 3: While you now explain that Aspartic acids were added, please explain why adding these negative charges is beneficial. Also, when you first describe peptide 12, please explain in which functional domain of the IAV PB2 this sequence is lying. Also, specify what is the corresponding sequence in influenza B.

An explanation as to why we chose the negatively charged peptide variant instead of the positively charged peptide variant is now added in the text (line 112-114). Briefly, the positively charged variants showed a similar antiviral effect but also encoded an aspecific toxic effects on human red blood cells, which would make future *in vivo* experiments impossible. Furthermore, the specific PB2 domain that is targeted by peptide 12 is now specified in the text (line 95): the cap-binding domain. Finally, as PB2 in influenza A and influenza B are structurally very similar, the corresponding sequence in influenza B is located on the same position in the cap-binding domain of PB2. This is now also highlighted in the text (line 261-262).

Point 5: The reviewer agrees with reviewer 2 that two-fold reduction in PFU (lung virus titer) is minimal, and it is very hard to believe that this effect was significant. If you want to keep the *in vivo* results, you should use much more subtle wording and explain that the *in vivo* effect was 'marginal'. [Editorially we think that it is okay that the synthetic amyloid is not quite ready for therapeutic demonstration *in vivo* and ask that you caveat and walk back this claim.]

Although we are confident that the effect observed is significant (n=13 mice over 6 independent experiments), we completely agree with the fact that our molecule is not ready for therapeutic demonstration. As was already highlighted in the text, we pinpointed this issue by showing the lack of *in vivo* stability. In the revised manuscript we now also included an extra sentence highlighting the need for more stable peptide variants to obtain improved therapeutic activity (line 141-143).